# Simultaneous dendritic voltage and calcium imaging and somatic recording from Purkinje neurons in awake mice

Christopher J. Roome [1] & Bernd Kuhn [1]

Spatiotemporal maps of dendritic signalling and their relationship with somatic output is fundamental to neuronal information processing, yet remain unexplored in awake animals. Here, we combine simultaneous sub-millisecond voltage and calcium two-photon imaging from distal spiny dendrites, with somatic electrical recording from spontaneously active cerebellar Purkinje neurons (PN) in awake mice. We detect discrete 1—2 ms suprathreshold voltage spikelets in the distal spiny dendrites during dendritic complex spikes. Spikelets and their calcium correlates are highly heterogeneous in number, timing and spatial distribution within and between complex spikes. Back-propagating simple spikes are highly attenuated. Highly variable 5–10 ms voltage hotspots are localized to fine dendritic processes and are reduced in size and frequency by lidocaine and CNQX. Hotspots correlated with somatic output but also, at high frequency, trigger purely dendritic calcium spikes. Summarizing, spatiotemporal signalling in PNs is far more complex, dynamic, and fine scaled than anticipated, even in resting animals.

[1] Optical Neuroimaging Unit, Okinawa Institute of Science and Technology Graduate University (OIST), 1919-1 Tancha, Onna-son, Okinawa 904-0495, Japan. Correspondence and requests for materials should be addressed to C.J.R. (email: chris.roome@oist.jp) or to B.K. (email: bkuhn@oist.jp)

Dendritic integration is fundamental to information processing within the brain. Extensive experimental studies in vitro and theoretical approaches have provided great insight into this complex process, yet determining how neurons decode synaptic inputs in awake animals remains an important and challenging endeavour in neuroscience[1–3]. Synaptic decoding occurs principally at the dendritic level, whereby complex geometries of dendritic processes, their nonlinear electrical properties and spatiotemporal distribution of their synaptic inputs facilitate rapid readout of thousands of inputs to perform precise input−output computations[4,5].

Receiving more synaptic inputs than any other neuron, cerebellar Purkinje neurons (PN) are among the most complex, in both geometry and functionality. A tremendous experimental[6–18] and computational effort[19–25] has been invested to unlock their functional properties. But due to the technical challenges of recording rapid (~ 1 ms) signals from fine (~1 μm) dendritic processes in vivo, fundamental functional properties, such as the spatiotemporal pattern of sub- and suprathreshold dendritic signals and their relation to somatic output, remain unknown.

Electrical recording from dendrites in vivo is challenging and often limited to anaesthetized animals[18] and restricted to single dendritic processes without somatic recording[26,27]. Conversely, optical functional imaging in awake animals using chronic cranial windows[28] and synthetic or genetically encoded indicators[29,30] provides high spatial resolution[3], but typically uses calcium indicators, that report only suprathreshold signalling at a temporal resolution limited by second messenger and indicator dynamics.

Thus, insofar as describing how dendritic activity relates to somatic activity, paired patch-clamp recordings from neurons in brain slices prove most insightful[11,15]. However, such preparations lack physiological inputs present in awake animals. A complete description of neuronal activity requires spatiotemporal recordings of both dendritic and somatic activity in awake animals.

Here, we explore single-neuron signalling multidimensionally, in its physiological environment with lowest possible disturbance of the system. To do this, we combined simultaneous dendritic voltage and calcium two-photon imaging with electrical somatic recordings, or drug application, from spontaneously active PNs of headfixed, alert but resting mice.

## Results

**Simultaneous awake dendritic voltage and calcium imaging**. Experiments were performed on headfixed mice, awake and sitting on a cylindrical treadmill (Fig. 1a, b). A chronic cranial window with access port[31], positioned over cerebellar lobule V, provided optical and physical access to single PNs (Fig. 1c). To record simultaneous dendritic voltage and calcium, PNs were intracellularly labelled with the synthetic voltage-sensitive dye ANNINE-6plus[32], and genetically encoded calcium indicator GCaMP6f[30] (Fig. 1d, e).

Dendritic imaging was performed using two-photon microscopy in linescan mode at 2 kHz (Fig. 1f, g; raw data shown in Supplementary Fig. 1). Linescan recordings were focused on fine spiny PN dendrites that are inaccessible with recording electrodes, at a depth less than 50 μm beneath the pia mater (Fig. 1e). Linescans (~ 200 μm in length) along PN dendrites recorded multiple dendritic branches within a single line. With a spine to dendritic shaft surface ratio of ~ 4:1 in mice[33] and homogeneous distribution of ANNINE-6plus in the membrane (attained more than 12 h after PN labelling), the dendritic voltage signals we detect originate primarily from dendritic spines, and to a lesser extent from dendritic processes. Similarly, spines have a significant contribution to calcium signals due to a spine to dendritic shaft volume ratio of 1.4:1 at this location[33].

Somatic activity of labelled PNs was simultaneously recorded using an extracellular recording electrode placed at PN soma, to detect the two characteristic forms of PN somatic output: simple spikes (SS) and complex spikes (CS)[6,34]. These signals were immediately recognizable by their well-characterized waveforms and their negative or positive going events respectively, and their average firing rates (SS frequency: $50 \pm 20$ Hz, CS frequency: $0.9 \pm 0.2$ Hz (mean ± s.d., 7 PNs)) (Fig. 1h).

PNs receive excitatory synaptic input from two distinct pathways: climbing fibres (CFs) and parallel fibres (PFs)[10]. We began by characterizing suprathreshold dendritic responses generated by CF input. Relaying information from inferior olive nuclei, a single CF makes approximately 500 synaptic contacts (in rat) with PN proximal dendrites[35–37] delivering a powerful membrane depolarization at ~1 Hz[6,18,34]. Accordingly, prominent CF-evoked dendritic voltage signals were readily identifiable in all linescans, at an average rate of $0.9 \pm 0.2$ Hz (mean ± s.d., 7 PNs) (Fig. 1f: black triangles). All CF-evoked events had concurrent dendritic calcium transients, with peak amplitude of $21.3 \pm 16.8\%$ $\Delta F/F$ (mean ± s.d., 7 PNs), and a somatic CS; as expected following CF input[8,9,15] (Fig. 1f−h).

We first averaged linescans spatially (across the full dendritic width) to measure the temporal dynamics of dendritic spikelets at 2 kHz temporal resolution. Voltage imaging detected rapid bursts of suprathreshold dendritic spikelets (here termed 'dendritic complex spikes' DCS) typically comprising 2−5 distinct spikelets with an average width (time from first to last spikelet) of $7.1 \pm 4.4$ ms (mean ± s.d., 7 PNs) (Fig. 2a). Individual dendritic spikelets were remarkably rapid and distinct (returning immediately to baseline). Mean peak $\Delta F/F$ for the initial spikelet was $15.8 \pm 3.3\%$ $\Delta F/F$ (318 spikelets, mean ± s.d., 7 PNs) corresponding to ~ 34 mV depolarization[38], similar in amplitude to the $35 \pm 4.3$ mV spikelets measured via patch pipette from primary and secondary PN dendrites in vivo[18]. Average full widths at half maximum (FWHM) of the initial spikelet within the burst was $1.4 \pm 0.2$ ms (318 spikelets, mean ± s.d., 7 PNs).

Dendritic calcium increased roughly incrementally with each DCS spikelet (Fig. 2a: see black arrows), in agreement with CF-evoked voltage and calcium transients being predominantly mediated by voltage-gated P-type calcium channels[18,39,40]. There was a clear correlation between the number of DCS spikelets and the resulting calcium peak (Supplementary Fig. 2) (Spearman's $\rho$ is 0.72, $p < 10^{-52}$). It is important to highlight, however, that the DCS width and corresponding calcium signals were strikingly variable from one DCS to the next in all PNs (population coefficient of variation (CV) for DCS width: $0.6 \pm 0.3$, and CV for peak calcium: $0.5 \pm 0.1$ (mean ± s.d., 7 PNs)).

We saw a wide range in peak calcium evoked by the same number of DCS spikelets, and considerable overlap of calcium elevations evoked by different DCS signals. This was most apparent for DCS bursts with >3 spikelets, revealing a nonlinear voltage−calcium relationship for these events (Supplementary Fig. 2b and d). We also found that while DCS initial spikelets had similar mean peak amplitudes between PN recordings, they were variable in amplitude with substantial deviation (~ 20%) from the mean within each PN (Fig. 2b). Spikelet amplitudes in the spiny dendrites ranged $15.8 \pm 3.3\%$ corresponding to $27−41$ mV[38], a similar range as found in the primary and secondary dendrites[18].

The DCS waveforms were faster than in previous recordings via intradendritic and patch pipette recording and lacked the secondary plateau potential following CF activation[7,18]. This is most likely due to differences in recording location. While the large smooth dendrites are typically targeted by recording electrodes, the ANNINE6-plus optical signal originates predominantly from fine spiny processes with highest membrane surface area. Our observations of rapid and temporally variable dendritic

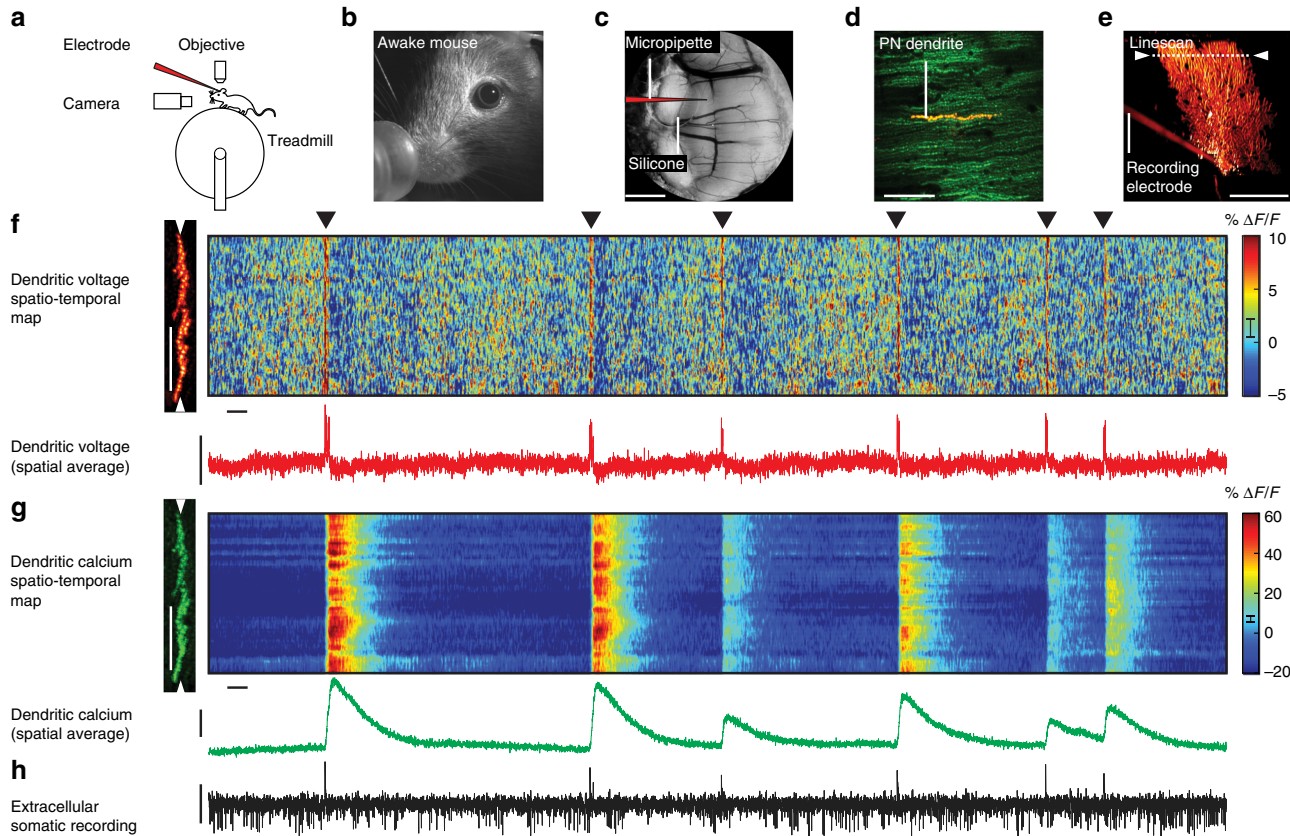

**Fig. 1** Simultaneous dendritic voltage and calcium imaging and somatic recording from Purkinje neurons in awake mice. **a** Experiments were performed on headfixed mice, alert and resting on a rotating treadmill. **b** A behaviour camera was used to monitor spontaneous movement in alert resting mice. **c** A cerebellar chronic cranial window with access port was used to label and record from single PNs in lobule V of the vermis. Scale bar 1 mm. **d** 2P image of distal PN spiny dendrites labelled with voltage sensitive dye ANNINE-6plus (red) and genetically encoded calcium indicator GCaMP6f (green), resulting in double labelling (yellow). Scale bar 100 μm. **e** Reconstruction of a single labelled PN showing position of 2P linescan (white arrows) and extracellular electrode placed at the soma. Scale bar 100 μm. **f** Spatiotemporal map obtained from 5 s of linescan recording (with 5 ms and 5 μm boxcar filtering) show spontaneous dendritic voltage signals. Filled triangles indicate dendritic complex spikes (DCS). Subthreshold dendritic voltage modulation is also visible in the spatiotemporal voltage map. Vertical scale bar 50 μm; horizontal scale bar 100 ms. Red trace shows spatially averaged voltage, recorded at 2 kHz resolution. Vertical scale bar 20% ΔF/F. **g** Corresponding spatiotemporal map for dendritic calcium. Vertical scale bar 50 μm; horizontal scale bar 100 ms. Green trace shows spatially averaged calcium. Vertical scale bar 20% ΔF/F. **h** Corresponding extracellular recording (black trace) showing somatic CS and SS events (positive and negative going events respectively). Vertical scale bar 100 pA. Error bars in colour maps show estimates for shot noise

spikelets, measured predominantly from spine membranes, indicate a remarkable potential for temporal precision of dendritic integration within spiny dendrites of PNs. Hence, distinct input signals (the spikelets) can be reliably generated and readily distinguished during dendritic integration; counted with millisecond precision, potentially at the level of a single spine.

All DCS events reliably generated corresponding somatic CS outputs, temporally interrupting regular SS firing, with an SS pause[41]. As with DCS events, somatic CSs were highly variable, particularly with regard to the SS pause (SS pause: $30 \pm 19$ ms, CV for SS pause: $0.6 \pm 0.2$ (mean $\pm$ s.d., 7 PNs)) (Supplementary Fig. 2c). The somatic SS pause is thought as an important output signal, generated by CF input[23,24] and has been shown to correlate with DCS signalling in vitro[15]. In vivo we saw a weak correlation between the number of dendritic spikelets and the corresponding SS pause (Spearman's $\rho$ is 0.19, $p < 10^{-3}$). A marginally stronger correlation was found between peak dendritic calcium and the corresponding SS pause (Pearson's $\rho$ is 0.32, $p < 10^{-9}$) (Supplementary Fig. 2e). In particular, this relationship did not agree with the strongest input signals (>3 spikelets), and relatively short SS pauses often followed the largest dendritic calcium signals. The most likely explanation for this weak

dendrite−soma relationship is that in vivo the DCS and SS pauses are influenced not only by CF input strength but also by background excitatory and inhibitory synaptic inputs, as previously anticipated[15].

In addition to DCS events, our paired PN recordings also identified single dendritic voltage spikes (DS) not triggered by CF input and that have not been detected previously in vivo (Fig. 2c: open triangle). These events occurred far less frequently than DCS events. Throughout a recording period of 340 s from seven PNs, we detected only 34 DS events, 50% of which were detected in a single PN and appeared to occur spontaneously. Compared to DCS events, DS events evoked much smaller calcium transients and had very different temporal profiles of dendritic voltage, showing a gradual ramping in membrane potential, terminating with a single, sharp spike. These events appear similar to those evoked in vitro by repetitive PF stimulation[17], extracellular glutamic acid iontophoretic application[42] or sustained dendritic depolarization[7]. Such events were predicted to occur in response to strong PF input onto spiny dendrites[12–14,43].

Somatic signals corresponding to DS events also varied in appearance from the CF somatic output signal (Fig. 2d). Initiation of somatic CSs always preceded the first DCS spikelet, consistent

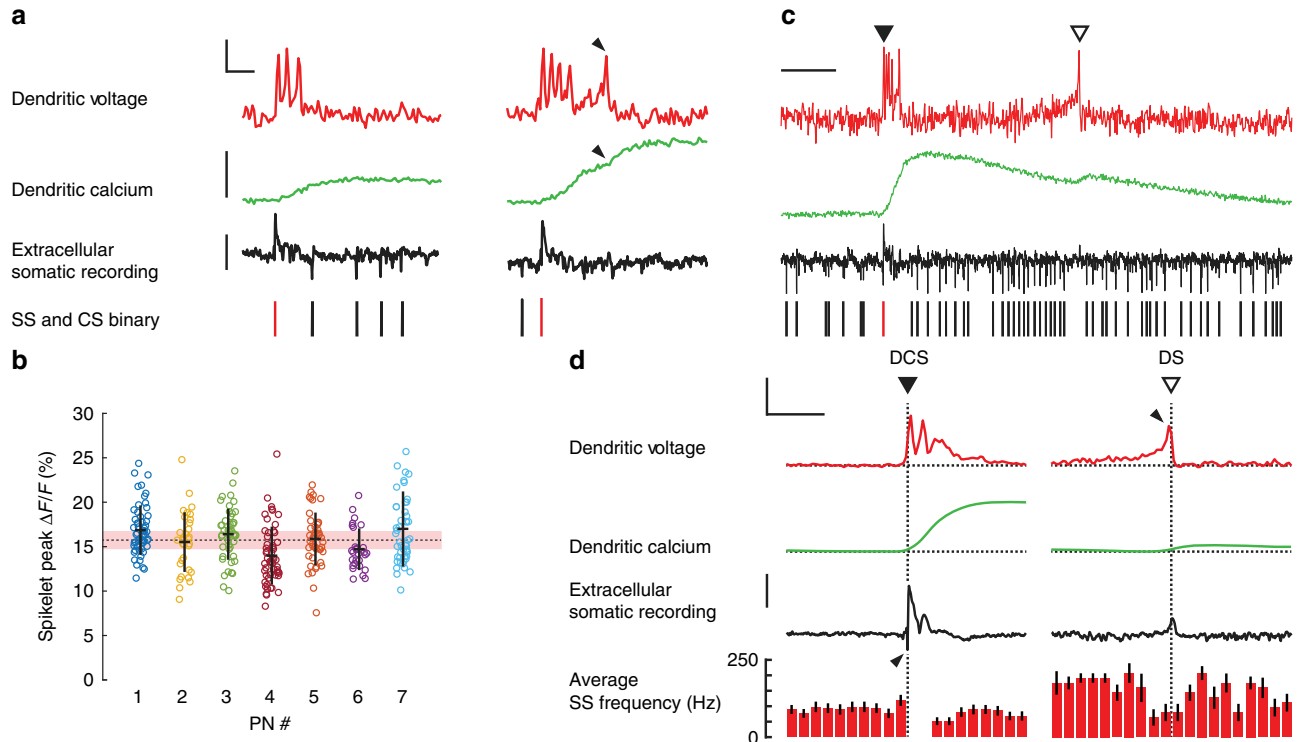

**Fig. 2** Two types of rapid and variable suprathreshold signals detected in spiny dendrites of awake mice. **a** Examples of dendritic complex spike (DCS) events (selected from Fig.1f) showing voltage (top red traces; vertical scale bar 10% ΔF/F, corresponding to about 21 mV; horizontal scale bar 5 ms), calcium (middle green traces; vertical scale bar 40% ΔF/F), and the corresponding extracellular somatic recording (bottom black trace; vertical scale bar 100 pA). Somatic binary signal showing SSs (black bars) and CSs (red bars). Black arrows indicate a late spikelet and a corresponding increment in calcium. **b** Amplitudes of the initial spikelets from all DCS recorded from 7 PNs (318 DCS in total), mean ± s.d. shown for each PN, group mean shown by dashed line and the pink bar shows error due to shot noise. **c** An example of a single dendritic spike (DS) event (open triangle) following a DCS event. Scale bar 50 ms. **d** Average dendritic voltage (vertical scale bar 10% ΔF/F; horizontal scale bar 25 ms), calcium (vertical scale bar 20% ΔF/F) and extracellular somatic recording (vertical scale bar 50 pA), and SS frequency (in 5 ms bins) from 49 DCS events (left) and 13 DS events (right) recorded from the same PN. Note the high SS firing rate surrounding (but not during) the DS events and the lack of a corresponding somatic action potential (downward going spike) in the somatic recording during DS events, in **c** and **d**. Black arrows indicate the onset of DCS and DS events. Bars show mean ± s.e.m

with the finding that somatic CSs are initiated near the soma and triggered by voltage-gated sodium channels[15,16,44]. Initiation of somatic CSs occurred 0.9 ± 0.2 ms (mean ± s.d., 7 PNs) prior to DCS initiation. In contrast, DS events had no corresponding somatic signal. Instead DS were initiated in the dendrites and occurred following high SS firing rates, comparable in frequency to recordings made in vitro[17].

It was apparent from our initial observations of suprathreshold dendritic signalling that the precise temporal pattern generated by DCS and DS events—their spikelet number and timing, and the resulting dendritic calcium signal—could not predict the corresponding somatic signal alone. This raised the question as to whether the variability of the dendrite−soma relation could be due, at least in part, to spatial inhomogeneity of dendritic signalling between dendritic processes.

**Spatiotemporal mapping of suprathreshold dendritic signals.** To explore this possibility, we divided dendritic linescans into four spatially equal segments (each ~ 50 μm in width). Each segment was averaged spatially and filtered temporally (1 ms boxcar). The resulting spatiotemporal voltage and calcium maps allowed us to resolve dendritic spikelets in response to CF input within four regions of the dendrite simultaneously (Fig. 3). As predicted, we saw high spatiotemporal variability between dendritic segments of both voltage and calcium DCS signals, evoked by the same CF input (Fig. 3b, c).

All segments of dendritic voltage showed bursts of rapid spikelets evoked by CF input, but their appearance differed between segments. During the DCS events the most active 'hot' dendritic segments had greater tendency for spikelet generation than the least active 'cold' segments. In accordance, corresponding calcium transients showed increased peak calcium elevation in hot segments than in cold segments, and time-locked increments in calcium elevation coincided with local dendritic spikelets in the voltage signal (see black arrows in Fig. 3b, c). Importantly, spatiotemporal patterns of both voltage and calcium signalling also changed dramatically from one CF input to the next, even during the same recording. We saw a similar range of spatiotemporal variability during DCS events in segments of all PNs (Supplementary Fig. 3).

To quantify DCS spatiotemporal variability we selected the strongest DCS signals from each PN for analysis (65 ± 20% of all DCS detected in 20 PNs). These events had calcium peaks >5% ΔF/F and at least two spikelets with amplitudes exceeding three standard deviations of the baseline voltage recording. For each DCS event the calcium peak and the number of spikelets was counted in each dendritic segment and compared. The hottest and coldest segments were defined as having the highest and lowest calcium peak, respectively. Sorting hot and cold segments in this way yielded an average hot segment calcium peak of 16.5 ± 6% ΔF/F and a corresponding average cold segment calcium peak of 10.4 ± 5% ΔF/F (mean ± s.d., $p = 1.4 \times 10^{-11}$, paired

two-tailed $t$ test, 20 PNs) (Fig. 3e). In agreement, the corresponding spikelet count was also significantly higher in hot segments compared to cold segments (Average hot segments spikelet count: 3.6 ± 0.4, average cold segments spikelet count: 2.6 ± 0.7 (mean ± s.d., $p = 1.5 \times 10^{-6}$, paired two-tailed $t$ test, 20 PNs)) (Fig. 3f).

Several earlier in vitro studies demonstrated that CF-evoked calcium signals show spatial variability[8,9,43] and more recently that CF-evoked calcium signals were graded by sensory stimulation[45].

Modulation of dendritic voltage through synaptic excitation and inhibition was proposed to underlie the spatiotemporal variability of CF-evoked calcium transient[18,45–47]. It is important to note however that several interplaying mechanisms could influence local differences in membrane excitability, whose interaction would explain the nonlinear voltage−calcium relationship that we observe in awake animals. Such mechanisms include intrinsic activity and stochasticity of dendritic ion channels, inhomogeneity in their distribution and local dendritic geometry[25,48,49], synaptic activation

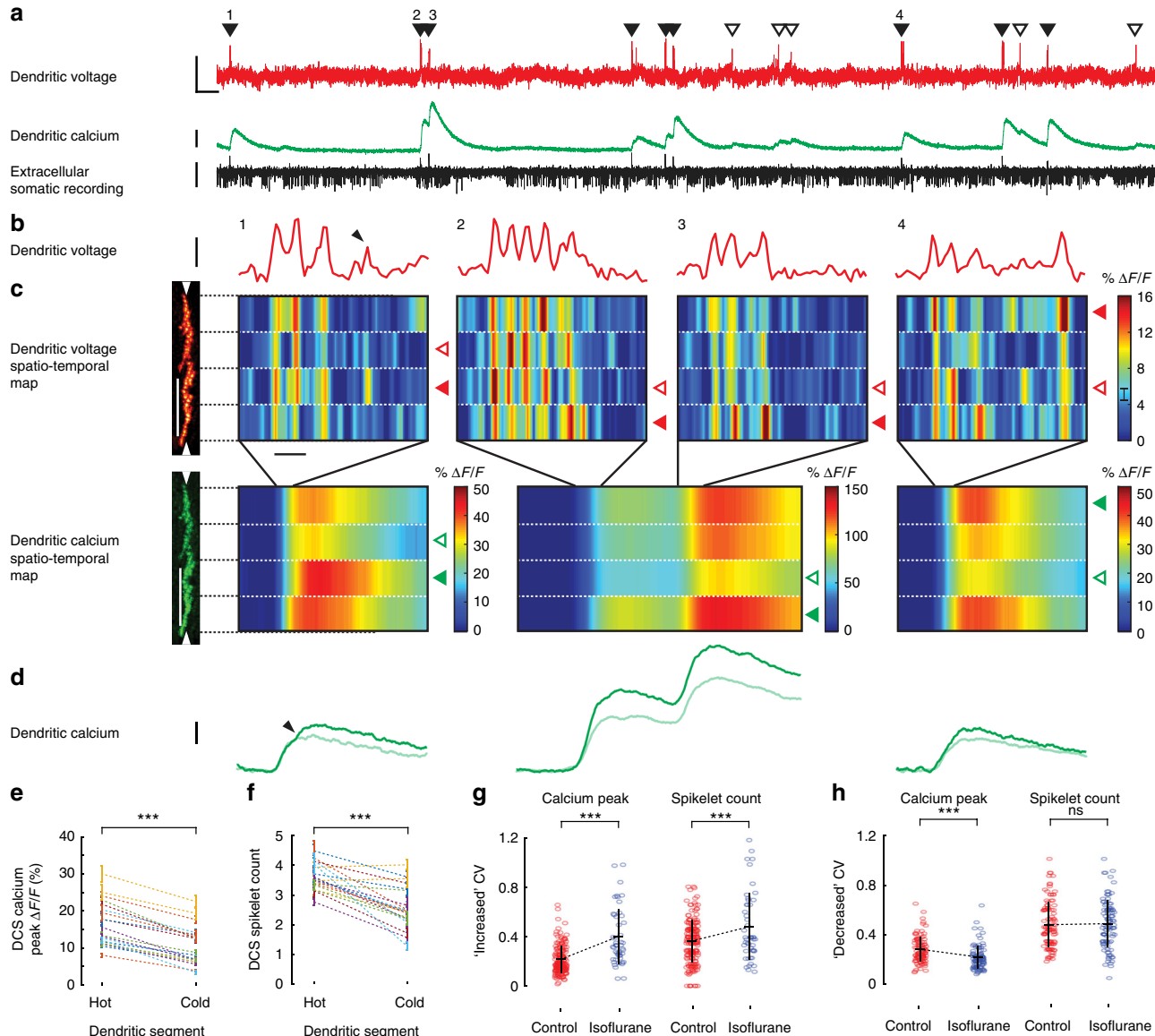

**Fig. 3** Suprathreshold dendritic complex spikes reveal high spatiotemporal variability in voltage and calcium signalling. **a** Simultaneous dendritic voltage and calcium and somatic recordings made from an awake mouse, with DCS and DS events indicated by filled and open triangles respectively. Voltage vertical scale bar 20% ΔF/F; calcium vertical scale bar 30% ΔF/F; extracellular somatic recording vertical scale bar 100 pA; horizontal scale bar 250 ms. **b** Four DCS examples (numbered in **a**) showing spatial average of dendritic voltage (red traces). Vertical scale bar 10% ΔF/F, corresponding to about 21 mV; horizontal scale bar 5 ms. **c** Corresponding spatiotemporal maps for voltage (vertical scale bar 50 μm; horizontal scale bar 5 ms), and calcium (vertical scale bar 50 μm; horizontal scale bar 50 ms) recorded from four dendritic segments (1 ms temporal filtering for voltage and 10 ms temporal filtering for calcium). Filled and open triangles (voltage; red, calcium; green) in all DCS examples indicate the hottest and coldest dendritic segments respectively, based on the peak calcium elevation and **d** shows the corresponding calcium traces from the hottest (dark green traces) and coldest (light green traces) segments. Vertical scale bar 20% ΔF/F. Black arrows in the first DCS example show a late voltage spikelet confined to a single dendritic segment in **b** and a corresponding calcium increment in (**c**) from the same dendritic segment. **e** DCS calcium peaks and **f** corresponding DCS spikelet counts in hot and cold dendritic segments, recorded from 20 PNs, paired two-tailed $t$ test. DCS coefficients of variation (CV) between all four dendritic segments, for calcium peaks and spikelet counts in awake (control) and anaesthetized (isoflurane) conditions. CV data has been grouped for PNs showing **g** increased CV or **h** decreased CV during anaesthesia, unpaired two-tailed $t$ test, ***$p < 0.001$, ns $p > 0.05$. Bars show mean ± s.e.m

of T-type calcium channels and mGluRs[47,50,51] and calcium release from internal stores[14,52–54].

To investigate the cause of the DCS spatiotemporal variability further, we compared DCS signals recorded from the PNs while the mouse was awake vs. while under anaesthesia (1% isoflurane) (Supplementary Fig. 4). It is expected that sensory input carried by PFs (and consequently the SS firing rate) is reduced during anaesthesia. In agreement we found that the average SS frequency was reduced during anaesthesia (Supplementary Fig. 4b) (SS frequency awake: $53 \pm 19$ Hz, SS frequency anaesthetized: $30 \pm 12$ Hz (mean $\pm$ s.d., $p = 0.03$, paired two-tailed $t$ test, 7 PNs)). The average DCS calcium peak was also reduced during anaesthesia (DCS calcium peak awake: $14.0 \pm 5.1\%$ $\Delta F/F$, DCS calcium peak anaesthetized: $9.3 \pm 4.8\%$ $\Delta F/F$, (mean $\pm$ s.d., $p = 0.01$, paired two-tailed $t$ test, 20 PNs)) (Supplementary Fig. 4a). DCS frequency, DCS spikelet amplitude and spikelet counts however did not change significantly (Supplementary Fig. 4c−e). Consequently, the fraction of DCS signals with calcium peaks >5% $\Delta F/F$ was reduced to $35 \pm 11\%$ during anaesthesia (Supplementary Figs 1, 4f).

We compared the spatiotemporal variability of DCSs in awake and anaesthetized conditions by calculating the CV of calcium peak and of spikelet counts between dendritic segments for each DCS (Fig. 3g, h). Anaesthesia had a modulatory effect on DCS CVs which varied between different PNs, even within the same mouse. Surprisingly, the majority of PNs (10 out of 15) displayed an increased CV in calcium peak (CV awake: $0.21 \pm 0.18$, CV anaesthetized: $0.40 \pm 0.23$ (pooled mean $\pm$ s.d., $p = 5 \times 10^{-14}$, unpaired two-tailed $t$ test, 10 PNs)) and also in spikelet count (CV awake: $0.36 \pm 0.11$, CV anaesthetized: $0.48 \pm 0.27$ (pooled mean $\pm$ s.d., $p = 7 \times 10^{-4}$, unpaired two-tailed $t$ test, 10 PNs)) during anaesthesia. The remaining PNs (5 out of 15) displayed a decrease in CV in peak calcium (CV awake: $0.22 \pm 0.10$, CV anaesthetized: $0.16 \pm 0.09$ (pooled mean $\pm$ s.d., $p = 3 \times 10^{-6}$, unpaired two-tailed $t$ test, 5 PNs)), but a similar trend was not present in spikelet count CV (CV awake: $0.42 \pm 0.18$, CV anaesthetized: $0.43 \pm 0.19$ (pooled mean $\pm$ s.d., $p = 0.7$, unpaired two-tailed $t$ test, 5 PNs)).

An increase in DCS spatiotemporal variability during anaesthesia was unexpected, but also proved that DCS spatiotemporal variability was not caused by movement artefacts during the linescan recording in the awake mouse, indicating a dependence of DCS spatiotemporal variability on the mouse behavioural state rather than a purely stochastic mechanism.

Using simultaneous dendritic voltage and calcium imaging at high spatial and temporal resolution, we confirm in the awake animal that spatiotemporal variability of the DCS calcium signal occurs due to differences in spikelet generation at different locations of the spiny dendrites. Surprisingly, variability within and between DCS is observed during spontaneous neuronal activity even while the mouse is resting.

Our recordings show that regions of spiny dendrites in PN of awake mice exhibit considerable spatiotemporal variability in response to CF input with high temporal precision. We confirm that different regions of spiny dendrites support a unique pattern of dendritic spiking influenced by local modulation of dendritic voltage, and thereby selectively fine-tune calcium signals at those locations.

**Relating subthreshold dendritic voltage to somatic activity**. We next sought to characterize subthreshold voltage modulation in the spiny dendrites and its relationship with somatic activity. Far outnumbering CF synapses, PN dendrites also receive numerous subthreshold excitatory inputs from cerebellar granule cells PF synapses[35]. In an impressive convergence of inputs, PFs synapse

with PN spiny dendrites at a very high density of ~ 3.4 spines per micron[33] to relay sensory information from all parts of the body and modulate PN SS output[7,55]. Given the high density of PF synaptic input at our recording location we anticipated that subthreshold modulation of dendritic voltage is predominantly driven by PF input, while stochastic activation of calcium channels may also contribute[48].

We began by characterizing the relationship between dendritic voltage and somatic SS activity (Fig. 4). Assuming spatially homogeneous back-propagation we spatially averaged linescans to determine the average voltage across all dendritic processes. Using SS triggered averaging (SS recorded at the soma) we detected subthreshold back-propagating SS signals in spiny dendrites (Fig. 4a). Previous in vitro studies using paired patch pipette recordings made on PN primary and secondary dendrites detected highly attenuated passive back-propagating SSs, due to dendritic filtering and the lack of dendritic voltage-gated Na$^+$ channels[11]. In agreement, our back-propagating SS signals occurred 0.5–1.8 ms (population mean: $1 \pm 0.4$ ms (mean $\pm$ s.d., 7 PNs)) after the SS recorded at the soma. They were highly attenuated in amplitude (peak amplitude: $0.6 \pm 0.3\%$ $\Delta F/F$; $1.3 \pm 0.6$ mV (mean $\pm$ s.d., 7 PNs)) and dramatically broadened in their time course (FWHM: $3.5 \pm 2$ ms (mean $\pm$ s.d., 7 PNs)). This signal attenuation is predominantly due to cable filtering effects of dendritic processes[22], but could also be actively dampened by voltage-dependent potassium channels in the dendrites[56] as they back-propagate from their site of origin, the axon initial segment[44], toward the most distal dendritic branches and spines. Consequently, the variety of SS waveforms that we measured between different PNs (Fig. 4a) is likely due to the unique morphological differences of each PN, their size and extent of dendritic branching (Supplementary Figs. 5-7).

Cross-correlation between average dendritic voltage and the SS binary trace was used to examine their temporal correlation (Fig. 4b). While the dendritic SS signal peak always occurred after the somatic SS signal (according to its somatic origin), there was a clear depolarization in dendritic voltage beginning ~10 ms before somatic SS onset. Indicating that at least on average, somatic SS are triggered following a subthreshold dendritic depolarization. Presumably this dendritic depolarization propagates passively towards the soma to evoke SS initiation, as we would expect from PF input[10,11].

To explore subthreshold voltage modulation further, we filtered linescan traces temporally to reduce optical noise (10 ms boxcar) (dark red trace in Fig. 4c). Comparing dendritic voltage with SS binary traces (binary trace in Fig. 4c), we saw a clear relationship between subthreshold dendritic voltage and the resulting SS firing rate, over longer timescales (~100 ms). This was confirmed by cross-correlation of the filtered dendritic voltage traces with SS binary traces, and we found no correlation between filtered calcium traces and the SS binary traces (Fig. 4d). An interesting contradiction to this rule however were the suprathreshold DCS and DS events, during which dendritic membrane potential increased rapidly beyond threshold while average SS firing rate temporally decreased (Fig. 2d), i.e. a sublinear relationship between dendritic and somatic activity.

We used convolutions of SS binary traces with the average dendritic SS signal to compare dendritic voltage with the dendritic voltage predicted from the back-propagating SS alone (blue trace in Fig. 4c, also with 10 ms temporal filtering). We found that the range in subthreshold dendritic voltage was significantly larger than predicted by the back-propagating SS convolution. As an alternative direct approach, we found pharmacological block of action potentials (APs) by 0.2% lidocaine and selective blockade of excitatory synaptic input by CNQX also reduced subthreshold voltage modulation (Fig. 4e) and recovered after washout

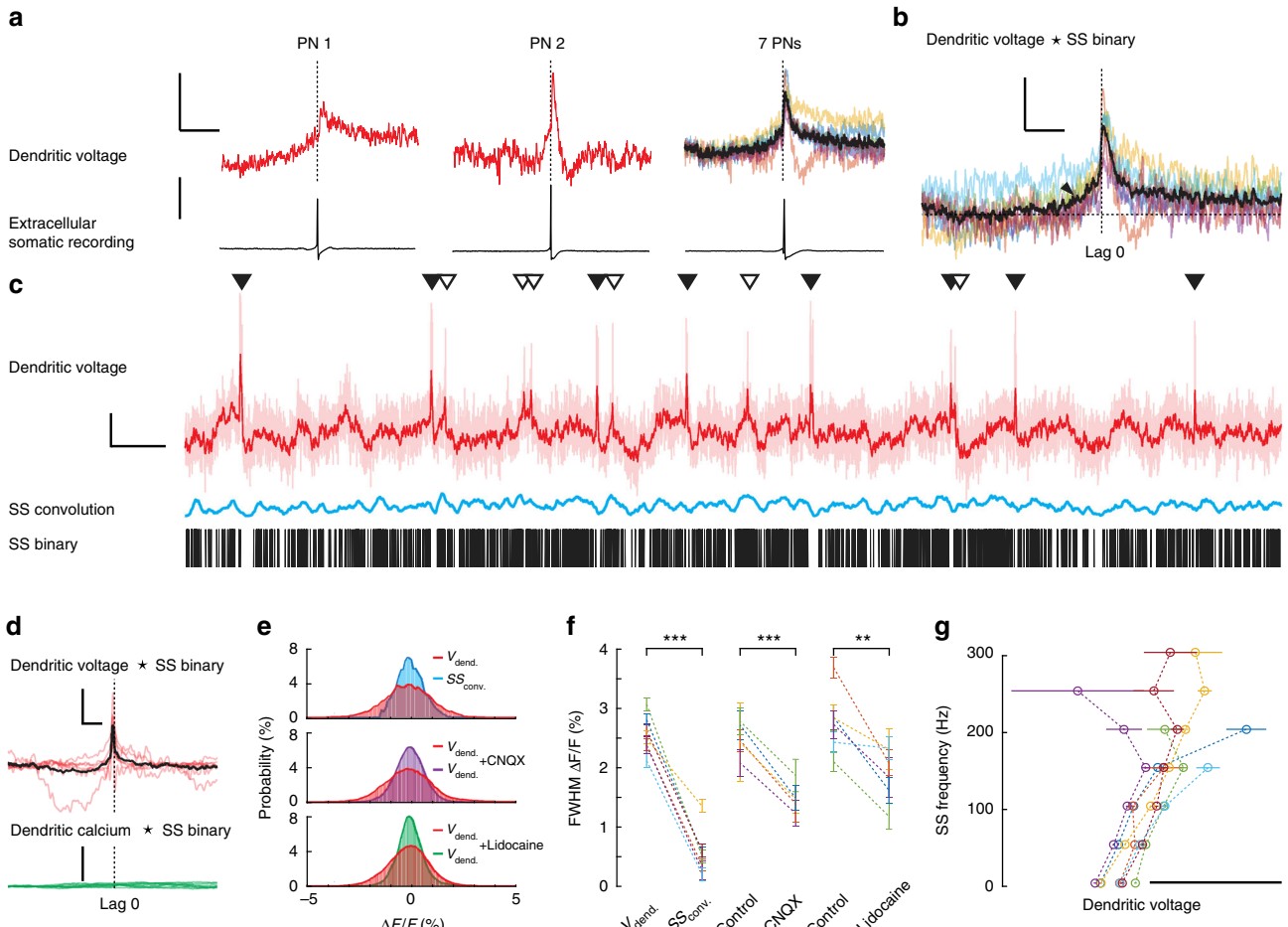

**Fig. 4** Subthreshold dendritic voltage reveals back-propagating action potentials and regimes of linear and nonlinear relationship with somatic activity. **a** Back-propagating action potentials (SS) recorded from PN spiny dendrites in awake mice, showing the most (left: 3614 averages) and least (middle: 1799 averages) attenuated signals recorded from different PNs, and signals from all seven PNs (right: black trace is group average). Voltage vertical scale bar 0.5% $\Delta F/F$, corresponding to about 1 mV; extracellular somatic recording vertical scale bar 50 pA; horizontal scale bar 20 ms. **b** Cross-correlation (CC) between SS binary signal (recorded at the soma) vs. spiny dendrite membrane potential. Black arrow indicates dendritic depolarization preceding SS initiation. Vertical scale bar 0.5% $\Delta CC/CC$; horizontal scale bar 10 ms. **c** 10 s of dendritic voltage recording, spatially averaged (light red trace). Dark red trace is dendritic voltage with 10 ms temporal filtering. Simultaneously recorded SSs are shown with binary (black bars), and corresponding convolution of back-propagating SS signal and SS binary (blue trace). Vertical scale bar 5% $\Delta F/F$; horizontal scale bar 500 ms. **d** Normalized cross-correlations between filtered dendritic voltage and SS binary traces (red traces, black trace is group average; vertical scale bar 3% $\Delta CC/CC$; horizontal scale bar 100 ms) and between corresponding dendritic calcium and SS binary traces (green traces; vertical scale bar 0.3% $\Delta CC/CC$). **e** Normalized probability distributions of dendritic voltage (red) and SS convolution voltage (blue) calculated using all recordings from PN shown in **c**, (top); of dendritic voltage (red) and dendritic voltage during CNQX application (purple) calculated from a single PN (middle); and of dendritic voltage (red) and dendritic voltage during lidocaine application (green) calculated from a single PN (bottom); DCS and DS events were excluded. **f** Full width at half maximum (FWHM) of the corresponding dendritic voltage probability distributions measured from individual PNs (5–7 PNs per group), bars show mean ± s.d. of shot noise, paired two-tailed $t$ test, ***$p < 0.001$, **$p < 0.01$. **g** Relationships between average dendritic voltage and somatic SS frequency recorded from seven PNs, calculated using a 20 ms kernel with 1 ms increments. Scale bar 5 mV. Bars show mean ± s.e.m

(Supplementary Fig. 8). Probability distributions of subthreshold dendritic voltages (i.e. excluding suprathreshold dendritic spikes) were compared with voltages predicted by the back-propagating SS convolution, and with subthreshold dendritic voltages recorded following the pharmacological manipulations (Fig. 4f). The FWHM of probability distributions were consistently broader for dendritic voltage than for SS convolutions, and for dendritic voltage following pharmacological manipulation (dendritic voltage FWHM: 2.6 ± 0.3% $\Delta F/F$; SS convolution FWHM: 0.6 ± 0.4% $\Delta F/F$ (mean ± s.d., $p = 1.9 \times 10^{-5}$, paired two-tailed $t$ test, 7 PNs)), (dendritic voltage control FWHM: 2.5 ± 0.3% $\Delta F/F$; dendritic voltage + CNQX FWHM: 1.5 ± 0.2% $\Delta F/F$ (mean ± s.d., $p = 1.2 \times 10^{-4}$, paired two-tailed $t$ test, 5 PNs)), (dendritic voltage control FWHM: 2.8 ± 0.5%

$\Delta F/F$; dendritic voltage + lidocaine FWHM: 1.9 ± 0.4% $\Delta F/F$ (mean ± s.d., $p = 0.009$, paired two-tailed $t$ test, 6 PNs)).

Relationships between subthreshold dendritic voltage and SS firing rate from seven PNs are summarized in Fig. 4g (using a 20 ms sliding kernel at 1 ms increments to calculate average dendritic voltage and corresponding SS firing rates). We found at low SS firing rates the dendrite−soma relationship was well defined and appeared approximately linear. Higher SS rates revealed their nonlinear nature however. In some PNs this was apparent through a sublinear relation of dendritic and somatic activity, as would be associated with the DCS and DS events or inhibitory synaptic input at the soma. Other PNs displayed a supralinear behaviour at increasing SS firing rates that could have

resulted from additional PF input located outside our imaging area.

Considering the strong attenuation of back-propagating SS signals, it is unlikely that this would be the predominant mechanism underlying spatial modulation of the DCS signal. The range in dendritic voltages far exceeded those generated by SS back-propagation alone (even at high firing rates), and we detected a clear dendritic depolarization, preceding SS initiation. Thus we conclude that the subthreshold dendritic voltage modulation we record is generated locally within the dendrites, and not due to passive SS back-propagation. A more probable explanation is that local subthreshold dendritic modulation is generated by the excitatory PF synaptic inputs and also the inhibitory synaptic input that the PN dendrites receive. This was confirmed through a reduction in subthreshold dendritic modulation by blockade of all APs and also by selective block of excitatory synaptic inputs. Given that subthreshold dendritic signals are generated within spiny dendritic processes by asynchronous excitatory and inhibitory synaptic input, we expect that they, like the suprathreshold signals, display spatiotemporal variability that reflects synaptic input patterns. We next sought to characterize the spatiotemporal pattern of subthreshold voltage modulation in the spiny dendrites.

**Spatiotemporal mapping of subthreshold dendritic signals.** The spatiotemporal pattern of subthreshold signalling in dendrites has not yet been described in vivo. To explore subthreshold voltage and calcium signalling in spiny dendrites and its relation to somatic activity, we used spatial and temporal filtering of linescans to create spatiotemporal dendritic voltage and calcium maps (boxcar widths 5 μm and 5 ms) (Fig. 5a, b, and also Fig. 1f, g). The 10 s spatiotemporal dendritic voltage maps revealed numerous rapid and spatially localized depolarizing signals (~ 100−200 events/50 μm; Fig. 5a). The depolarizing events, which we call 'hotspots', were inhomogenously distributed across dendritic processes showing highly variable activity rates and spatial distribution, with regions of low and high frequency activity (Fig. 5a, see insets and white arrows).

A similar signal pattern was not observed in corresponding calcium spatiotemporal maps (Fig. 5b, see insets and white arrows). The $\Delta F/F$ we detected during hotspot events far exceed that estimated by our shot-noise analysis (shot-noise error bar is shown in Fig. 5a colour scale) and their inhomogeneous spatiotemporal pattern also would not be expected from shot noise. Interestingly, only in surrounding DS events did we see a reliable association between hotspot events and dendritic calcium signals. In this case, there was a clear increase in hotspot activity prior to and immediately following the dendritic calcium spike (see insets and open triangles in Fig. 5a, b, voltage and calcium maps).

To analyse hotspot events quantitatively and to investigate their relationship with somatic activity, we selected hotspot regions of interest (ROIs) from voltage maps using a lower threshold limit of 5% $\Delta F/F$ (corresponding to a voltage change of about 11 mV) and, to further eliminate events potentially caused by optical noise, we removed hotpixels (see Methods) and selected only hotspot events that were >3 ms in duration and >1.5 μm in spatial width (i.e. >3 neighbouring pixels). From the resulting ROI mask (Fig. 5c) we estimated the fraction (%) of hot dendritic processes caused by hotspot events (Fig. 5d). There was a clear association between the % of hot dendrite and somatic SS firing; high SS firing rates occurred during higher percentages of hot dendrites (see black boxes in Fig. 5d), further supporting the biological nature of hotspot events. In the example of ROI mask shown, DCS and DS events are included to show the stark contrast of dendritic spatial extent between localized hotspot

events and widespread suprathreshold events that activate the majority of dendritic processes.

To investigate the origin of the hotspot signals we repeated hotspot detection on linescan recordings in control (awake) conditions and following application of 0.2% lidocaine (Fig. 5e−h) or 100 μm CNQX (Fig. 5i, j), and during isoflurane anaesthesia (Fig. 5k, l). Hotspot dynamics and activity was characterized by selecting hotspot events detected within a 50 μm dendritic segment from all spatiotemporal maps from each PN (avoiding gaps between dendritic branches).

We found the average frequency and area of individual hotspot events were significantly reduced following pharmacological manipulation by lidocaine (hotspot frequency control: 6.6 ± 3.7 Hz, hotspot frequency lidocaine: 5.3 ± 3.2 Hz), hotspot area control: 40.4 ± 87.4, hotspot area lidocaine: 22.6 ± 23.0 (pooled mean ± s.d., $p < 0.001$, unpaired two-tailed $t$ test, >3000 hotspots in each group from three PNs) and also by CNQX (hotspot frequency control: 7.0 ± 4.9 Hz, hotspot frequency CNQX: 4.6 ± 3.5 Hz, hotspot area control: 31.0 ± 75.5, hotspot area CNQX: 20.9 ± 16.4 (pooled mean ± s.d., $p < 0.001$, unpaired two-tailed $t$ test, >4500 hotspots in each group from five PNs)). Anaesthesia also reduced hotspot frequency and area in some (but not all) PNs (hotspot frequency control: 7.1 ± 4.6 Hz, hotspot frequency isoflurane: 6.2 ± 4.1 Hz, hotspot area control: 35.0 ± 69.4, hotspot area isoflurane: 29.1 ± 54.8 (pooled mean ± s.d., $p < 0.001$, unpaired two-tailed $t$ test, >18,900 hotspots in each group from 13 PNs)). Thus we concluded that the hotspot signals are generated by both excitatory synaptic input and stochastic voltage fluctuations within the dendrite, with the larger area hotspot events resulting predominantly from synaptic input.

To investigate hotspot temporal and spatial dynamics we selected isolated hotspot events (~ 3% of the total hotspot events detected per PN) and averaged in temporal and spatial dimensions, using the unfiltered spatiotemporal maps to confirm that hotspot dynamics were not an artefact of filtering. Isolated hotspots were defined as having no overlapping hotspot ROIs in their near vicinity (±30 ms, ±5 μm of the ROI centre). Temporal and spatial profiles of isolated hotspot events from a single PN are shown in Fig. 6a and sorted by their temporal duration. Temporal and spatial hotspot profiles averaged from all PNs are shown in Fig. 6b.

Isolated hotspot events were rapid and highly localized. The average duration (at threshold) of isolated hotspot events was 5.5 ± 0.8 ms (mean ± s.d., 7 PNs) and their average width was 4.4 ± 0.7 μm (mean ± s.d., 7 PNs). Only after extensive hotspot averaging did we detect a corresponding signal in the green channel spatiotemporal map (see black arrows in Fig. 6b). Since these signals were identical in time course to hotspot voltage signals, but of much smaller amplitude, they are voltage signals in the green spectral range of ANNINE-6plus fluorescence, and not calcium signals. The small amplitude can be explained by mixing with GCaMP6f fluorescence and a low voltage sensitivity of ANNINE-6plus in the green spectral range[32,57]. This voltage signal with different amplitudes in different channels further confirms that the hotspots are voltage signals and not noise artefacts.

Probability histograms were used to summarize dynamics of all hotspot events detected in all PNs (Fig. 6c−f). The average duration and spatial extent of all hotspot events was 6.8 ± 6.6 ms and 5.7 ± 3.7 μm respectively (mean ± s.d., 7 PNs). The average amplitude of all hotspot events was 6.8 ± 1.0% $\Delta F/F$ (mean ± s.d. between PNs), (shot-noise error: ±2% $\Delta F/F$), corresponding to about 14 mV, and we calculated the average frequency of hotpot events to be 6.6 ± 2.4 Hz/5 μm (mean ± s.d., 7 PNs).

It is important to note that beyond DCS events the probability of detecting large regions of synchronously active dendrites was

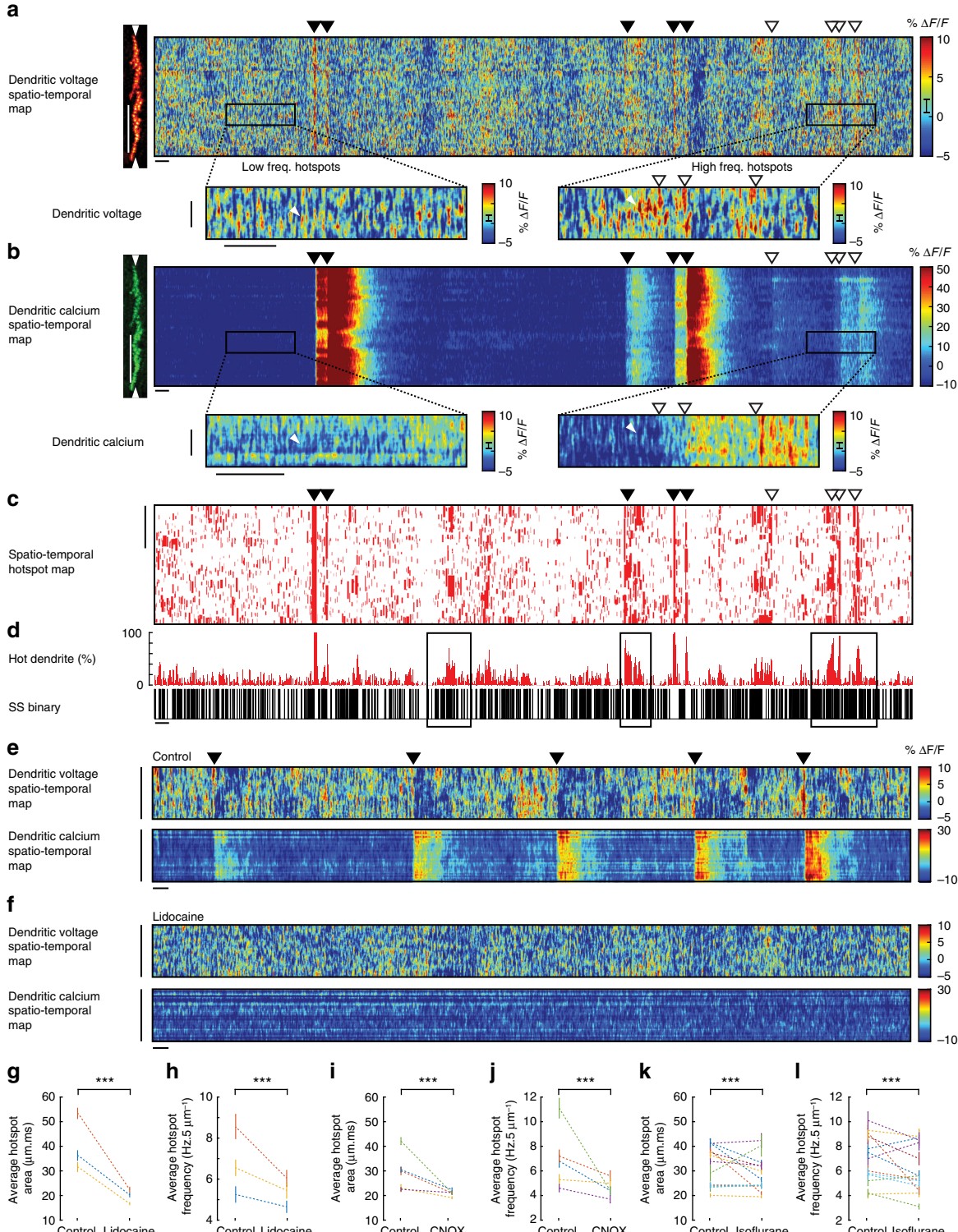

**Fig. 5** Voltage imaging reveals the spatiotemporal patterns of subthreshold hotspots in PN spiny dendrites of awake mice. **a** Spatiotemporal maps for voltage and **b** calcium recording from PN spiny dendrites. Vertical scale bar 50 μm; horizontal scale bar 100 ms. Inset boxes show examples of low and high hotspot activity in the voltage recording and the corresponding calcium recording (inset vertical scale bar 5 μm; horizontal scale bar 100 ms), white triangles indicate single hotspot events detected in the voltage map, but not in the calcium map. **c** Corresponding hotspot mask (hotspot ROIs from **a**) showing spatiotemporal hotspot map. Vertical scale bar 50 μm. **d** Percent of hot dendrite (5 ms bins) calculated from (**c**), and corresponding somatic SS binary trace (black bars), black boxes show epochs of high hotspot frequency and high SS frequency. Filled and open triangles throughout indicate DCS and DS events respectively. Horizontal scale bar 100 ms. **e** Spatiotemporal maps for voltage and calcium during awake (control) conditions and **f** following application of 0.2% lidocaine. Vertical scale bar 50 μm. Horizontal scale bar 100 ms. Average hotspot area and frequency in control conditions and following pharmacological manipulation by **g**, **h** lidocaine (3PNs), **i**, **j** CNQX (5 PNs), and **k**, **l** during anaesthesia by isoflurane (13 PNs), paired two-tailed *t* test, ***p < 0.001. Bars show mean ± s.e.m

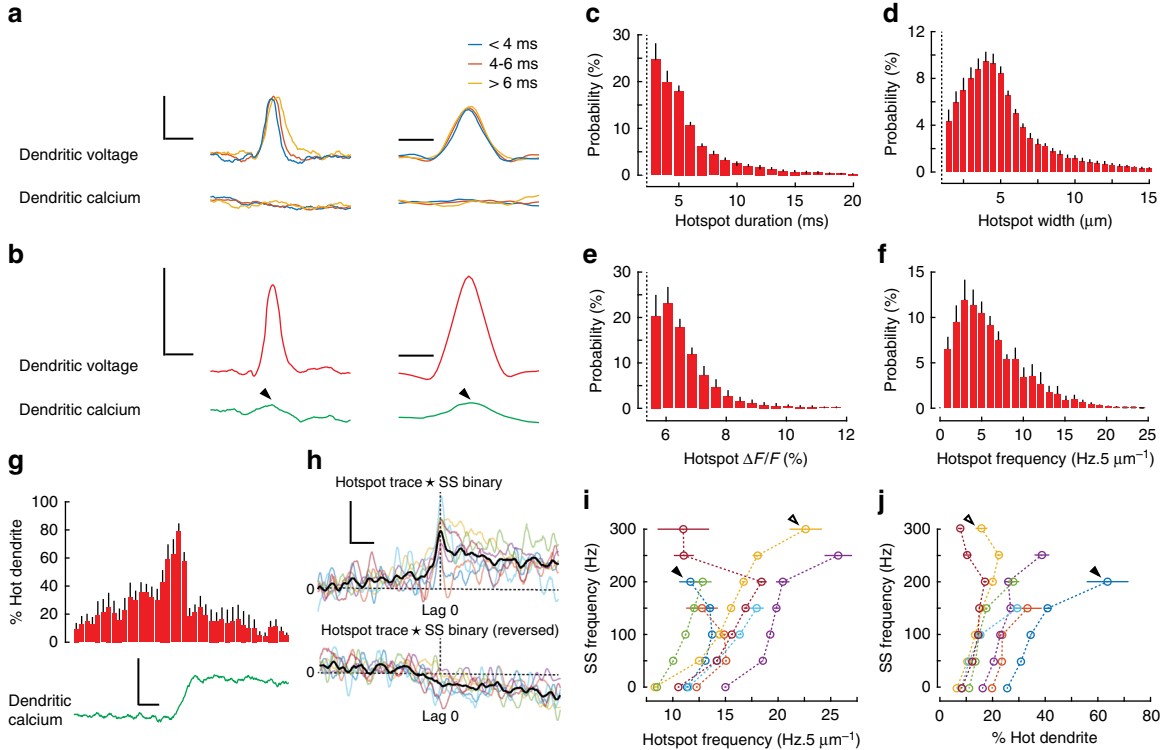

**Fig. 6** Rapid and localized subthreshold dendritic voltage hotspots show nonlinear relationships with dendritic calcium and somatic output. **a** Averaged isolated hotspots (216 in total, single PN) showing profiles of voltage (top traces) and calcium (bottom traces) (vertical scale bar 5% ΔF/F, corresponding to about 11 mV) in the temporal dimension (left traces; horizontal scale bar 10 ms) and the spatial dimension (right traces; horizontal scale bar 5 μm), grouped based on temporal duration of their ROIs. **b** Averaged isolated hotspots (1534 in total) showing profiles of voltage (top traces) and calcium (bottom traces) from all seven PNs (vertical scale bar 5% ΔF/F, corresponding to about 11 mV) in the temporal dimension (left; horizontal scale bar 10 ms) and the spatial dimension (right; horizontal scale bar 5 μm). Probability distributions for **c** the duration, **d** width, **e** amplitude and **f** frequency of all detected hotspots from seven PNs. **g** Average hotspot activity (percent hot dendrite in 5 ms bins) and average dendritic calcium (green trace; vertical scale bar 5% ΔF/F; horizontal scale bar 20 ms) during 13 DS events from a single PN. **h** Cross-correlations from all PNs, showing temporal relationship between hotspot traces and SS traces (top coloured traces), and cross-correlations from the same PNs, using reversed hotspot input traces (bottom coloured traces), thick black traces show group averages (vertical scale bar 0.1% ΔCC/CC; horizontal scale bar 20 ms). **i** Relationships between hotspot frequency (Hz/5 μm) and SS frequency for seven PNs and **j** the corresponding relationship between percent hot dendrite and SS frequency, calculated using a 20 ms kernel with 1 ms increments. Bars show mean ± s.e.m

very low (Fig. 5c). DS events were notable exceptions. During DS events we saw a nonlinear increase in the percentage of hot dendrites due to hotspot activity, preceding DS initiation, which evoked a calcium signal (Fig. 6g). A similar nonlinear relationship between PF-induced EPSPs and evoked calcium signals in spiny dendrites was described in vitro[13], for which it was predicted that 20−30 PFs need to be activated in the local dendritic area to evoke a calcium transient.

To explore the temporal relationship between hotspot events and somatic SS rate, we created hotspot input traces for the temporal hotspot pattern by convoluting the hotspot temporal profile (Fig. 6b) with hotspot counts in the spatiotemporal hotspot map (see Supplementary Fig. 9 for example). We then performed cross-correlations between hotspot input traces and corresponding SS binary signals on 500 ms trace segments containing no DCS or DS events. Average cross-correlations were calculated for each PN, and all cross-correlations showed a sharp peak in correlation coefficient centred at lag 0 (Fig. 6h). Reversing the same hotspot traces and repeating the cross-correlation with the SS binary signal (as a negative control) eliminated the correlation (cross-correlation at lag 0 ($\Delta C/C_{lag0}$): 0.17 ± 0.01% and reversed $\Delta C/C_{lag0}$: −0.04 ± 0.04%, (mean ± s.d., $p = 0.0004$, paired $t$ test, 7 PNs)), thus confirming the temporal relationship between hotspot events and SS activity.

Relationships between dendritic hotspot and somatic SS activity were determined using a 20 ms sliding kernel to calculate the hotspot frequency (per 5 μm), the percent of hot dendrite caused by hotspot activity, and the corresponding SS output rates. Relationships between hotspot frequency and somatic SS frequency are shown in Fig. 6i, and the corresponding relationships between the percentage of hot dendrite (caused by hotspot activity) and somatic SS frequency are shown in Fig. 6j. A nonlinear relationship was found in both cases. Again these relationships approached linearity at lower SS frequencies. Interestingly, in some PNs, the behaviour of these two relationships appeared complementary at high SS frequencies (see arrows in Fig. 6i, j), whereby a sublinear relationship between hotspot frequency and SS activity appeared balanced by a supralinear increase in the percent of hot dendrites and vice versa. A summary for dendritic and somatic PN signalling in alert resting mice is shown in Table 1.

## Discussion
Our data explores theories of dendritic-somatic interplay, inspired through decades of experimental and computational investigation. We combined the voltage-sensitive dye ANNINE-6plus, the genetically encoded calcium indicator GCaMP6f and

**Table 1 Summary of supra- and subthreshold dendritic signals and somatic activity of PN in awake mice**

| PN # | Recording time (s) | Dendrite area ($10^4$ μm$^2$) | Dendrites | | | Soma | |
|---|---|---|---|---|---|---|---|
| | | | Superthreshold | | Subthreshold | | |
| | | | DCS freq. (Hz) | DS freq. (Hz) | Hotspot freq. (Hz/5 μm) | SS freq. (Hz) | CS freq. (Hz) |
| 1 | 50 | 1.6 | 1.12 | 0.06 | 11.3 | 41.2 | 1.12 |
| 2 | 50 | 2.2 | 0.66 | 0 | 7.7 | 40.0 | 0.66 |
| 3 | 50 | 1.5 | 0.98 | 0.32 | 5.8 | 85.3 | 0.98 |
| 4 | 50 | 1.8 | 1.12 | 0.12 | 4.0 | 56.0 | 1.12 |
| 5 | 50 | 1.8 | 0.92 | 0.02 | 7.0 | 49.1 | 0.92 |
| 6 | 50 | 1.7 | 0.58 | 0.16 | 5.1 | 22.3 | 0.58 |
| 7 | 40 | 1.6 | 1.23 | 0.08 | 5.3 | 55.4 | 1.23 |
| Mean ± s.d. | | 1.7 ± 0.2 | 0.9 ± 0.2 | 0.1 ± 0.1 | 6.6 ± 2.4 | 49.9 ± 19.4 | 0.9 ± 0.2 |

electrophysiology to simultaneously record spatially resolved spontaneous sub- and superthreshold dendritic voltage and calcium signals and somatic outputs from single neurons for the first time in an awake animal. While ideal recordings would sample activity from all neuronal processes simultaneously, our 2 kHz two-photon linescan imaging, focused on the most distal spiny dendritic processes of PNs, allowed voltage and calcium recording at high spatiotemporal resolution, from many dendritic processes simultaneously. Our technique exposed underlying complexities of electrical signalling in dendrites, invisible to calcium imaging techniques and our recordings offer a first sampling of the variability of dendritic integration in neurons of the same type in an awake animal. Dendritic voltage and calcium signalling of PNs in awake animals is remarkably complex, even while at rest, perhaps reflecting the motor tonus coordination of posture, expected to being handled in the vermis of the cerebellum.

Notably, we detected rapid (5–10 ms) subthreshold hotspot voltage signals, localized to fine dendritic processes with no corresponding calcium signal. Hotspots were partially blocked by AMPA/kainate antagonist (CNQX) and showed regimes of linear and nonlinear relationship with somatic SS rate. Averaging subthreshold dendritic voltage signals across dendritic processes also revealed back-propagating SSs, whose rising phase preceded axonal SS initiation, as expected from the integration of many overlapping PF-evoked EPSPs. The hotspot events we detect are similar in appearance to those predicted by computational models simulating PF-evoked EPSPs, which extended beyond spines with rapid temporal dynamics, activating fine dendritic branches[19–21].

We detected hotspot events in the finest dendritic processes, composed predominantly of spine membrane at PF synapses. With an average spatial width of 5 μm and an imaging depth of ~ 5 μm (Supplementary Fig. 10), this area of dendrite would hold ~ 15 spines in mice[33]. However, due to the limited spatial extent of our two-photon linescan imaging technique, this estimate for the spatial extent (in depth) of hotspot events may be underestimated. Additionally, we hypothesize that our size estimate for isolated hotspots is a result of morphological constrains, such as the cross section of dendritic branchlets with spines (estimated diameter of 5 μm) and a short branchlet length with many bifurcations. These constrains may explain the discrepancy between our measurement and the expected length constant[19–21].

In our imaging volume we detected an average hotspot frequency of ~ 7 Hz, or ~ 0.5 Hz per spine. Similar low frequency PF input would be expected from measurements of granule cell activity in vivo, which fire at a rate of 0.5 ± 0.2 Hz[58]. Thus, we propose the dendritic hotspot events are AMPA-mediated EPSPs, evoked by PF synaptic input, that do not elicit corresponding calcium signals, and

support earlier predictions that relatively few PF inputs are required to drive PN simple spike activity in vivo[55].

Such signals would be expected at PF synapses, since their EPSPs are predominantly mediated by AMPA receptors with very low calcium permeability[10,59]. In PNs P-type calcium channels are the primary source of calcium influx, requiring high voltage depolarization for activation i.e. during dendritic calcium spikes[14,40] (as in Fig. 5a). Subthreshold calcium signals in PN spiny dendrites following PF stimulation in vitro have been reported previously[12,13], but we could not detect them in resting mice. Interestingly, calcium signals evoked by sparse PF input were localized to spine heads and mediated by calcium release from internal stores[14]. Since we did not detect subthreshold calcium signals in alert but resting mice, even after extensive averaging, we speculate that such events might occur during specific behavioural states or during motor learning.

We did however detect nonclimbing fibre evoked dendritic spikes (DS) that occurred following a sharp rise in hotspot activity in the spiny dendrites. Unlike CF-evoked dendritic spikes, DS events generated a small elevation in dendritic calcium and had no directly correlating somatic activity (Figs. 2d, 6g). These suprathreshold dendritic voltage spikes had not yet been observed in vivo, but have been predicted to occur during elevated PF activity during sensory stimulation[45]. We also detected clustered regions of high frequency hotspot activity (Fig. 5a inset). These findings support predictions that clustered PF input onto PN dendrites could function as a potent stimulus for synaptic plasticity during learning at PF-PN synapses, through local activation of suprathreshold dendritic calcium spikes[18,45,60].

Importantly, we showed that spikelets evoked during DCS and DS events were rapid, discretely countable and spatiotemporally variable in the spiny dendrites. We demonstrated how fine-scale spatiotemporal spikelet variability, triggered by a formerly assumed monolithic CF input (see ref.[61] for discussion) is uniquely modified, based on the spatiotemporal map of dendritic voltage. In this regard the spatiotemporal maps of background synaptic activity direct the spread of CF input into hot dendritic processes, boosting local calcium influx. Inhibitory inputs would conversely shunt calcium influx in other processes[18,46]. Thus by favouring dendritic processes with coincident PF input, dendritic coincidence detection can be performed that has previously been shown to induce synaptic plasticity at these synapses[14].

More generally, we have revealed the extent of spatiotemporal variability of dendritic signals in vivo and their complex relationship with somatic output. Operating under distinct sub- and suprathreshold regimes, dendritic integration allows fine-scale timing and location-dependent computation of synaptic inputs

that impact somatic output over various timescales, while providing under specific circumstances, compartmentalization of dendritic branches as isolated computational units, operating independently of the soma, with no direct interference on somatic activity.

## Methods

**Animals and surgery**. All animal procedures were conducted in accordance with guidelines of the Okinawa Institute of Science and Technology Institutional Animal Care and Use Committee in an Association for Assessment and Accreditation of Laboratory Animal Care (AAALAC International)-accredited facility. Cerebellar chronic cranial window surgeries were performed on 2-month-old male C57/BL6 mice, using a 5 mm glass cover slip with silicone access port[31]. The window was positioned to allow imaging within lobule V of the cerebellar vermis and the access port was positioned to allow access to the imagining area via a micropipette for PN labelling and electrical recording. Silver wires were attached to the skull using wire glue and dental cement on either side of and anterior to the cranial window for EEG recording.

**Microscope setup**. We used a custom-built combined wide-field, two-photon microscope (MOM, Sutter Instruments) with either a ×5/N.A. 0.13 air objective (Zeiss) or a ×25/N.A. 1.05 water immersion objective with 2 mm working distance (Olympus) with ScanImage software[62]. Bright field imaging was performed using a sCMOS camera (PCO.edge, PCO). A femtosecond-pulsed Ti:sapphire laser (Vision II, Coherent), circularly polarized and under-filling the back focal plane of the ×25 objective, was used to excite fluorescence (typical laser power at 1020 nm: 60 mW), which was detected by two GaAsP photomultiplier tubes (Hamamatsu) in the spectral range of 490–550 nm (green: GCaMP6f) and 550–750 nm (red: ANNINE-6plus).

The mice were headfixed on a platform that consisted of a vertically rotating treadmill, head-plate stage and micromanipulator tower (Sutter Instruments), all mounted on a horizontally rotating stage (8MR190-90-4247, Standa). The sCMOS camera and infrared light source were used to record behavioural activity during recording at 37.3 fps. A second infrared video camera (Sony) was used to monitor mouse behaviour throughout the experiment. An EEG amplifier (Sigmann Elektronik) was mounted on the platform and connected to silver wires attached to the mouse skull during surgery to monitor EEG activity during awake and anaesthetized conditions. The micromanipulator (M-285, Sutter instruments) was used for GCaMP6f virus (UPenn Vector Core) and ANNINE-6plus dye (www.sensitivefarbstoffe.de, Dr. Hinner and Dr. Hübener Sensitive Dyes GbR) injection and also for electrophysiological recording.

**AAV injections**. One week following surgery mice were anaesthetized (1−2% Isoflurane) and head mounted for two-photon guided injection of the adeno-associated viral vectors (AAVs) into the PN layer approximately 150 μm below the dura. For this beveled quartz electrodes (0.7 mm ID, pulled and beveled to 10−20 μm tip diameter) containing AAV1.hSyn.Cre (2E13 GC/ml), AAV1.CAG.Flex. GCaMP6f (1.3E13 GC/ml), and 50 μM FITC in PBS at a ratio of 1:1:1 were used to specifically target PNs and visually control the position and size of the viral injection, while <0.1 PSI pressure was used to inject the virus for 1 min. After virus injection the pipette was retracted and the mouse was returned to its cage.

**Single neuron labelling with ANNINE-6plus**. One week after virus injection, GCaMP6f-expressing PNs were targeted for voltage-sensitive dye (ANNINE-6plus) single-cell labelling by electroporation, guided by two-photon microscopy. GCaMP6f-expressing PN were electroporated using a patch pipette containing 3 mM ANNINE-6plus dissolved in ethanol. Borosilicate glass (patch) pipettes with 1 μm tip diameter (7−10 MΩ) were used for electroporation and a stimulus protocol of 50 negative current pulses (−30 μA), 1 ms in duration at 100 Hz were delivered. Neutral pressure was applied to the patch pipette to prevent leakage of the dye/ethanol solution into tissue and the pipette was retracted immediately after the cell was loaded and replaced for further single-neuron labelling.

Typically, three PNs were filled per mouse on the same day, and after loading PNs with dye, mice were returned to their cages to allow the dye to spread to distal dendrites and throughout the entire cell. ANNINE-6plus is highly lipophilic so dye diffusion can take several hours (>12 h). After ~20 h, the brightest labelled cell was selected for imaging experiments, which were performed the day following PN labelling. This also guaranteed that the PN was healthy and had not been damaged by the labelling procedure. Where possible, several PNs were labelled in the same mouse (up to 5) and used for simultaneous dendritic voltage and calcium imaging recordings. Between neuron labelling sessions and following surgery, mice were returned to their cages and allowed to recover.

**Simultaneous dendritic voltage and calcium imaging**. Approximately 20 h after ANNINE-6plus labelling, extracellular recordings were performed on double labelled PNs with simultaneous imaging from distal dendrites (<50 μm below dura) in linescan mode at 2 kHz sampling rate. Labelled PNs were clearly visible in both

red and green channels indicating successful labelling with ANNINE-6plus and GCaMP6f. Mice were allowed to sit awake on the treadmill for at least 1 h before beginning the experiment. During recordings, mice were alert and headfixed sitting on a rotating treadmill. Bidirectional linescans, 512 pixels in width, lasting 10.5 s were performed at a line rate of 2 kHz. To limit photo-damage during linescans and to improve signal-to-noise ratio, the objective collar was rotated to elongate the excitation volume predominantly in the z-direction to ~5 μm (Supplementary Fig. 10). During the 10.5 s, no bleaching was observed (Supplementary Fig. 1a). Fine corrections in linescan orientation (with respect to PN dendrites) were done prior to the experiment using the rotating stage, on which the mouse treadmill and micro-manipulators were placed. The linescans measuring 256 μm in width were carefully positioned as superficial as possible as to include the full dendritic width of the most distal dendritic spiny PN branches, thus maximizing the total membrane area covered by the linescan, typically less than 50 μm below the pia mater. ANNINE-6plus is purely electrochromic showing linear responses across the full physiological voltage range and is well suited for recording neuronal membrane potential[31,63], with a temporal resolution limited only by the fluorescence lifetime (6.2 ± 0.1 ns in cortical tissue, see 'Fluorescence lifetime measurement of ANNINE-6plus' below). The femtosecond-pulsed Ti:sapphire laser was used to excite fluorescence at 1020 nm, near the red spectral edge of absorption[38]. To confirm optimal ANNINE-6plus sensitivity near the red spectral edge of absorption[38] and the mechanism of voltage sensitivity[31,64], different excitation wavelengths were tested (Supplementary Fig. 1d and e). Excitation near the red spectral edge of absorption to optimize voltage sensitivity allows for long-term simultaneous voltage and calcium dendritic recordings at least 500 s per recording session (Supplementary Figs. 6, Supplementary Fig. 1e) at different dendritic depths (Supplementary Fig. 7). As ANNINE-6plus is relatively hydrophobic compared to other voltage-sensitive dyes for intracellular application, the labelling lasts for at least 2 weeks (Supplementary Fig. 6). Due to an extended excitation point spread function (~1 × 1 × 5 μm³, Supplementary Fig. 10), used to increase the signal-to-noise ratio, the voltage signal is the average membrane potential in this volume encompassing spines and dendritic shaft.

**Electrophysiology**. After verifying voltage and calcium signals from the labelled PN, extracellular recordings were made from the same PN while mice remained awake, using quartz electrodes (0.7 mm ID, Sutter Instruments), beveled at a blunt angle of 40°, with a tip resistance of 7–10 MΩ. Extracellular recording electrodes contained 50 μM Alexa Fluor 594 Hydrazide (Invitrogen Molecular Probes: A10438), dissolved in 0.9% NaCl to allow visually guided positioning of the electrode tip onto the PN soma. To help reduce tip blockage and prevent dye leakage into the tissue, electrode tips were filled with 1−2% agarose gel (A9793, Sigma-Aldrich) containing 50 μM Alexa and 0.9% NaCl in water. For these recordings it was not necessary to apply pressure to the electrode, which could be inserted directly through the silicone access port in the cranial window, with the bath electrode placed above the window under the objective with saline as immersion medium.

Electrophysiological current recordings were made using a headstage and amplifier at a sampling rate of 10 kHz (EPC10, HEKA). An audio speaker connected to the HEKA amplifier output was used to aid positioning of the electrode onto the labelled PN soma by listening for simple spike and complex spike discharge. Once the electrode was positioned on the PN soma, a short (2 s) linescan was performed, simultaneously recording electrical activity from soma and optical signals from the dendrite. If coincident complex spike activity at soma and dendrite (calcium and voltage) was observed the experiment begin. Each linescan recording lasted 10.5 s and was initiated while the mouse was stationary on the treadmill, but alert, using the video camera in night mode to continuously monitor its behaviour.

In all experiments, recording began while the mouse was awake. From each PN we collected >40 s of awake recording in 10 s epochs, during which time we saw no photo-bleaching of ANNINE-6plus. After at least 40 s of data had been collected and well before any photo-damage was observed (maximum recording time for a single PN was about 4 min), mice were lightly anaesthetized and a z-stack image reconstruction was made with the extracellular recording electrode still in place to reconstruct the labelled PN. In 15 PNs (7 with somatic recording and 8 without) recordings were repeated while the mouse was anaesthetized (1% isoflurane) (Supplementary Fig. 1) and 3D reconstructions of labelled PNs were made (Fig. 1e, Supplementary Figs. 5-7).

**Pharmacology**. In a separate set of experiments (11 PNs), the extracellular recording electrode was replaced by a micropipette used for pharmacological manipulation. In this case, the micropipette: a beveled quartz micropipette (0.7 mm ID, Sutter Instruments) was placed ~50 μm below the PN soma and the drug (0.2% lidocaine (Sigma); to block APs, or 100 μM CNQX disodium salt (Tocris) dissolved in saline; to block excitatory synaptic input), was applied by pressure injection (<0.5 psi) for 10 min prior to imaging and reduced to <0.1 psi while dendritic voltage and calcium recordings were repeated in the awake mice. Dendritic recordings were repeated ~24 h after drug application (guaranteeing drug washout) to confirm that the labelled PN was not physically damaged by the drug application (Supplementary Fig. 8).

**Data analysis**. Linescan TIFF images were initially cropped in width using ImageJ (US National Institute of Health) to contain only the dendrite from the labelled PN using the red channel as a guide, and to eliminate green signals originating from neighbouring PNs. All subsequent data analysis was performed using custom-written programs with Matlab (MathWorks). Single unit electrophysiological recordings were analysed using Spike2 spike detection software (CED) to create a binary trace for SS (and to eliminate spikes originating from neighbouring PNs). All CS (occurring at ~1 Hz) were easily identified by eye, which could be confirmed by comparing simultaneous electrophysiology, voltage and calcium traces. CS binary traces were created using the initial sodium spike of the somatic CS to mark its onset. Noncomplex spike dendritic (calcium) events were also observed and a binary trace for these events was created where the onset of the calcium transient marked the event onset. All binary traces, raw images and electrophysiology were then imported into Matlab for analysis. Full linescan traces were imported into Matlab and interpolated (from 2 to 10 kHz) to compare with the electrical recording.

To analyse dendritic spiking events, the binary traces for SS, CS, and dendritic spike events were used to segment and align linescan traces to short time intervals, centred on each event. Short segments of linescan data were then analysed individually (thus minimizing movement artefacts) and could also be averaged between all events (e.g. SS triggered averaging) or spatially across dendritic segments or the entire dendritic width.

To calculate $\Delta F/F$ of either full linescans or segments, the green channel image was first scaled (using Matlab regress function) and fit to the red channel and then subtracted from the red channel. This removed crosstalk from the green channel. Average baseline red channel fluorescence was then subtracted from the time-varying red channel fluorescence and the result was divided by the average baseline of red channel fluorescence to give $\Delta F/F$ for the red channel. The $\Delta F/F$ calculation for the green channel was made in the same way as for the red channel, but it was not necessary to first subtract red channel fluorescence from the green channel as the voltage signal was neglectable compared to GCaMP6f. Instead, we measured the florescence intensity ratio of ANNINE-6plus baseline fluorescence of the green and red channel by labelling PNs with ANNINE-6plus only (no GCaMP6f fluorescence). The contribution of ANNINE-6plus baseline fluorescence could then be subtracted to scale the GCaMP6f $\Delta F/F$ signals accordingly. Relative fluorescence changes imaged with an excitation wavelength of 1020 nm were converted with a factor of 2.1 mV/% to estimate voltage changes[38].

**Shot-noise analysis and hotpixel removal**. Shot noise was calculated using a linear regression to fit mean intensity vs. standard deviation from each pixel in the linescan over the 10 s recording. Using the linear fit, relative noise was scaled depending on the spatial and temporal filtering. To minimize the impact of shot noise on hotspot detection 'hotpixels' within spatiotemporal maps with fluorescence exceeding 50% $\Delta F/F$ in red channel were removed prior to filtering and replaced with the mean $\Delta F/F$ for the entire trace (close to zero). Lateral movements (typically <5 μm) in linescan were also corrected using a cross-correlation of each linescan line (in the spatial dimension) with the average of all lines. Low frequency fluctuations in fluorescence intensity (<1−2 Hz) were subtracted.

**Behaviour analysis**. Behavioural recordings were made while the mouse was awake but sitting on a treadmill. A behaviour camera (sCMOS), capturing frames at 37.3 Hz, was used to record movements synchronized with the imaging and electrophysiology recordings.

**Fluorescence lifetime measurement of ANNINE-6plus**. To measure the fluorescence lifetime of ANNINE-6plus, cerebral cortex tissue in 0.9% NaCl saline was labelled with ANNINE-6plus (40 μM final concentration) and excited with a pulse-picked frequency-doubled ultrafast Ti:sapphire laser (MaiTai, Coherent) at 450 nm. The time-resolved fluorescence spectrum was measured with a streak camera (C10627, Hamamatsu) and analysed with IgorPro (WaveMetrics Inc.) by fitting a first-order exponential decay to the emission profile between 550 and 750 nm.

**Data availability**. We will make all raw data and Matlab codes available upon request.

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

## Acknowledgements

We thank Peter Fromherz (Max Planck Institute of Biochemistry, Martinsried, Germany) for the gift of ANNINE-6plus, the GENIE Program and the Janelia Research Campus for distributing GCaMP6f, Kieran Deasy for technical support, Andrea Giovannucci for good advice, Steven Aird for technical editing and Sam Wang for helpful feedback on the manuscript. We are grateful for the generous support and funding from the Okinawa Institute of Science and Technology Graduate University.

## Author contributions

C.J.R. designed the study, built the setup, collected and analysed data, and wrote the paper. B.K. designed the study, built the setup, and wrote the paper.

## Additional information

**Competing interests:** The authors declare no competing interests.

