## [Peer Review File · Nature Communications]

Reviewers' comments:

Reviewer #1 (Remarks to the Author):

In this manuscript, Roome and Kuhn examine the integrative properties of Purkinje cell dendrites in awake mice. Using a technically awesome approach combining simultaneous dendritic Ca²⁺ and voltage imaging as well extracellular electrophysiological recording at the soma, they report several key details of Purkinje cell dendritic electrogenesis that are quite novel. Their observations include: 1) Spatio-temporal variance of climbing fiber evoked spiking and Ca²⁺ activity within a dendrite. 2) Dendritic spikes that occur independent of the soma through the putative activity of parallel fibers. 3) Subthreshold depolarizing potentials generated by putative parallel fiber activity that is correlated with simple spike output at the soma. These data extend current knowledge of Purkinje cell dendritic function and therefore should be of wide interest both to the cerebellar field as well as those interested in properties of dendritic integration. Enthusiasm for the work is limited by a number of concerns that should be straightforward to address.

1) Purkinje cell activity measurements were performed in resting mice (awake and but not behaving). Therefore, all methodological detail regarding monitoring behavior that isn't pertinent to the experiments performed should be removed or clarified (e.g., neural measurements were obtained during periods of inactivity as determined by...). Otherwise, inclusion of such detail is confusing. For example, it is stated in the Methods that EEG recording was performed but there is no mention of EEG activity in the Results. Another example is whisker tracking measurements in Supplemental Figures 1 and 4 that are completely unrelated to the data presented nor are these measurements referred to in the text (as a side note, recordings were performed in an area of the cerebellum not known to be related to whisking).

2) The dendritic spike associated with the "startle" response is unconvincing. It occurs several hundred ms after the stimulus. Furthermore there is no reproducibility of this response. The "startle" DS spike is, apparently, only observed in one trial from one animal. Lastly, the authors present no evidence for a "startle" reflex (EEG?). It is advised that this data should be removed as it adds little to the manuscript. Perhaps this could be developed in future work.

3) The quantification and statistics that support claims regarding differences in "hot" and "cold" dendrites is not provided (i.e., more spikelets, higher spikelet peaks, spikelet spacing; lines 233-235). Relatedly, group data presented in Supplemental Figure 5 doesn't show a statistical difference in the average amplitudes of Ca²⁺ transients between "hot" and "cold" dendritic segments. Lastly, the authors conclude that variability in DCS signals occur due to spikelet generation and timing differences (lines 264 and 265) however there doesn't seem to be any quantification to support this claim.

4) The authors review possible factors that may contribute to spatio-temporal variability of the climbing fiber Ca²⁺ transient in the Results (Lines 268-276). This paragraph seems better placed in the Discussion section because there are no follow-up experiments that

require elaboration of these possibilities.

5) Do the putative parallel fiber evoked dendritic spikes show similar spatio-temporal variability in the dendrite as those evoked by climbing fibers? This should be quantified and reported for both the voltage and Ca²⁺ signals. That the DS spike appears widespread and not constrained to a branch or region is potentially very interesting. Does the widespread nature of the DS spike reflect propagation from a single initiation point in a specific branch due to the clustered activity of nearby parallel fiber inputs or does it result from the dendrite-wide activity of distributed but co-active inputs as inferred from the non-linear increase in percentage of hot spots shown in Figure 6g? This point should be clarified.

6) Cross correlations between "slow" dendritic activity (hills and valleys) and simple spike rates (lines 342-343) aren't shown.

7) If the isolated, subthreshold voltage hotspots are AMPA EPSPs as speculated in the Discussion, aren't they expected to decay passively along the dendrite? If so, how does their spatial domain (5-10 μ m) compare to the expected length constant of the Purkinje cell dendrite?

8) Quantification of high spiking rates and percent of "hot" dendrite (lines 418-419) is not provided.

9) In the legend of Figure 3, "d" is not included. It is probably mislabeled as "c".

10) "DCS" is first used in line 129 but isn't defined until lines 148-149.

Reviewer #2 (Remarks to the Author):

In this technically oriented paper, the authors recorded 2P-excited fluorescence (1020 nm) of a linescan, scanned as superficially as possible but estimated at < 50 μ m below the pia mater, through the distal dendritic trees of 7 Purkinje Neurons. The neurons are expressing GCaMP6f and have been electroporated with ANNINE-6plus. An extracellular electrode is positioned close to the soma of the transfected/electroporated cell and an extracellular recording is made in parallel with the optical recording. The excited fluorescence is spectrally split and recorded on two detectors (490-550 nm and 550-750 nm), the interpretation being that one detector records ANNINE fluorescence and the other GCaMP fluorescence. The ANNINE trace is corrected for bleedthrough of the GCaMP fluorescence by subtracting off a scaled version of the GCaMP trace. The authors provide interesting hypotheses on the relation between dendritic and somatic activity in awake mice, but in general more work needs to be done before bolstered conclusions can be drawn.

- The biggest player in dendritic heterogeneity, which is formed by the molecular layer interneurons providing inhibitory input to the Purkinje cells, is not taken into consideration. Please address at the experimental level.

- All records were focused on distal dendrites, as shown in Fig S2. The distal dendrites are unlikely to be innervated by climbing fibers; thus their responses indicate mainly the products of dendritic filtering rather than local CF inputs. Did the authors compare the responses from different parts of the Purkinje cell dendrites?

- The authors claimed that the dendritic complex spike signals were variable. Using line scan in awake mice is prone to be affected by motion artifacts, especially the distortion in z axis. To what extent the dendritic complex spike variability is explained by instable recordings is unclear. In general, detailed analysis of recording stability is essential; please address this as well as the details highlighted below.

- The dataset appears to be somewhat over-interpreted at this point. This is exacerbated by the lack of a proper noise analysis. The 3 lines devoted to shot noise in the supplementary information makes me wonder what the authors mean by signal and by noise. An example of this is the claim in line 462: average amplitude of hotspot events is 6.8 ± 1.0 % DF/F mean \pm SD, shotnoise 2% DF/F. I may misinterpret the notation but this seems to say that the measured noise is smaller than the theoretically minimal noise.

- The measurement is a linear line-scan through a set of dendrites. Interpretations of the effect of different timing in different part of the scan appears to be done without any correction/discussion of the actual difference in propagation length/time from the recorded points to branching points in the dendrite or to the soma, which raises doubts on the concept of relative timing of the signals in the different parts of the scan.

- Figure 2b: variability in spikelet amplitude seems to follow a normal noise distribution as expected for an optical recording with limited photons. The interpretation in the text is however that there is an underlying physiological phenomenon (line 170-171). Without proper noise analysis this conclusion cannot be supported.

- In lines 172 to 179 an interpretation of the data is provided that emphasizes the superiority of the optical recording over previous efforts, but it is noted that an expected secondary plateau potential is missing. This could be a signal to noise issue, but it could also mean that the recorded traces do not accurately reflect voltage waveforms. Without a proper noise analysis and most importantly without any direct measurement comparing a known voltage signal to the optical recording, it is not possible to interpret the data in a meaningful way.

- This reviewer wonders to what extent the authors can distinguish voltage-modulated fluorescence from non-voltage-modulated fluorescence. I wonder whether they can interpret a DF/F in one mouse as a spikelet amplitude and compare it to DF/F and spikelet amplitudes in another mouse, as levels of background fluorescence/tissue auto-fluorescence and non-voltage sensitive fluorescence are going to be different. Yet this is exactly what is done, which makes all conclusions drawn about relative magnitudes of spikelets suspect. A background correction is applied, but this can make things worse. For example, in figure S5 the difference in amplitude between spikelets of hot and cold segments is emphasized. The

voltage trace is obtained by subtracting a scaled version of the GCaMP trace. The undershoot after the spike in the cold segment makes it look as if simply a slightly wrongly scaled version of the calcium trace was subtracted, rendering the meaning of the resulting difference in spikelet amplitude uncertain. Given the small difference in spikelet amplitude and the nonlinearity in voltage response of ANNINE-6plus, it is also hard to say whether spikelet sizes of different mice can be compared, as no measure of baseline voltage is available.

- The authors give Kuhn 2004 as a reference though this deals with ANNINE6 rather than ANNINE-6plus; given the similarity in Fromherz 2008 of the spectra, and the fact that the authors base voltage sensitivity on Kuhn 2004, I assume the same nonlinearity holds too, but please address the issue in the discussion.

- In figure 4 an average SS is calculated and convoluted with the SS binary trace. Much is made of the fact that this trace shows less variation in DF/F than the actual trace of dendritic voltage. The averaging has effectively filtered out one source of noise in that shape, i.e. the spike to spike variation. It is therefore not surprising that figure 4d and e look like they do. Other traces of random noise treated the same way may show similar behavior. Please address.

- In figure 1f and g, their spatiotemporal maps are filtered with a 5 um boxcar spatial filter and a 5 ms temporal filter. Effectively this means that they re-sampled their data to 5 um and 5 ms (disregarding some minor effects of the actual filter shape), limiting their detection bandwidths to $0.2/\mu\text{m}$ and 200 Hz. They then interpolate the data again at 10 khz, but this does not change the fact that they filtered all higher frequency information out of their signal. This may invalidate their hotspot analysis. They select hotspots > 3 ms and $1.5 \mu\text{m}$, which effectively means single pixel noise at these bandwidths. This means that for instance the average temporal size of isolated hotspots (5.5 ± 0.8 ms) and average spatial width ($4.4 \pm 0.7 \mu\text{m}$) are hard to interpret, as they're looking at their detection bandwidth. The meaning of Figure 6a is therefore uncertain. The authors aim to show that crosstalk in the green channel shows a voltage signal, but that spike only means that the noise is not detector noise and does not exclude any common mode noise, i.e. in the laser intensity or detection electronics or even biological noise. Averaging random noise peaks, centered specifically at the maximum amplitudes of said noisepeaks, may give the same effect that the authors claim proves the voltage nature of their signal.

- The meaning of Figure 6h is unclear. The authors filtered out the DCS and CS events from their hotspot trace, which are also missing from the SS binary. A convolution of those traces would, based only on these artificially created correlated events alone, already show a peak around 0 lag, which would disappear with time reversal of one of the traces.

- The authors base a lot of their discussion and interpretation on the spatial range of their line-scan and the average number of dendrites and spines in the thus defined imaged lines that they expect from literature. I would argue that, for a paper that purports to spatially and temporally disentangle voltage dynamics in these subcellular regions, this is not a best practice.

- Linelengths are defined as ~ 200 μm or 256 μm ; at 512 pixel lines, this changes the number of pixels in a 5 μm boxcar filter, or the number of pixels in the lengthscale that is claimed as significant for hotspots. They further state that they have done a large part of their analysis on ROI, where they cropped out areas without dendritic membrane. This last action would presumably throw off the average number of dendritic spines in a volume, changing the discussion. It is not clear with relation to what space these average numbers have been defined. They further claim their focal spot is 1 μm and show an image of their focus that was made by scanning a 1 μm bead through the laser focus. This gives an imaged spot that, according to their scale bar, is also 1 μm . A 1 μm focus convoluted with a 1 μm bead would however not give a 1 μm spot, so either their focus is of a significantly different size than they think, throwing off their estimates for spine and dendrite numbers in the focal volume, or the spatial range of the experiment is off. Please address this issue.

Reviewer #3 (Remarks to the Author):

The manuscript by Roome and Kuhn leverages an impressive combination of challenging neurophysiological techniques to assess synaptic integration in cerebellar Purkinje neurons of awake mice. Despite decades of investigation into the mechanisms of dendritic function, few studies have been able to address how dendrites influence input processing in vivo, particularly in awake animals. Those few studies have generally employed Ca^{2+} imaging, which is an indirect report of dendritic voltage. Here, the authors combine Ca^{2+} imaging with voltage-sensitive dye imaging to quantify the organization of subcellular voltage changes in distal dendritic branches as a function of somatic output. The methodological approach is remarkable: the authors label single neurons with voltage sensitive dye via two-photon guided electroporation after an earlier surgery to infect many neurons with GCaMP ; they allow the dye to spread, and then reintroduce a pipette to record juxtacellular signals from the neuron with the VSD. The ability to index dendritic signals in both the voltage and Ca^{2+} domains by the somatic output pattern is tremendously useful (and impressive). While I have two experimental concerns (listed below), a major issue with the manuscript is in regards to the impact of the presented data. It's unclear what the findings of the study are, and what they mean for the field. The work is highly descriptive; as a result, I cannot quite figure out how this should change our thinking for how cerebellar dendritic integration works. The authors do not frame their results in the context of testing a hypothesis, or disambiguating prior controversies, or uncovering novel features of Purkinje neuron function. In fact, they mention that what they see in vivo matches what has previously been described in slices during direct dendritic recording. As it stands, it is hard to know what the manuscript is trying to say, aside from demonstrating that this kind of recording is possible. Therefore, its significance for the field is limited.

Regarding the technical approach, the use of voltage dyes often presents serious experimental caveats both in terms of altering the properties of the recorded neurons and interpreting the resulting signals. The authors cite previous studies of this particular dye, but there is very limited physiological quantification in these papers. Given how much work

the authors invest in their labor and time intensive in vivo experiments, a separate set of slice experiments to rigorously evaluate the dye seems necessary to me. Otherwise, it is unclear what the VSD signals recorded in vivo represent. Ideally, the authors would measure the linearity of the dye across a range of voltage amplitudes and kinetics from a structure in which they have a ground-truth patch-clamp electrical recording (i.e. the soma or a proximal dendrite), as well as critically evaluate any potential disturbances in the neuron's membrane properties produced by loading with the dye.

A further experimental concern pertains to the extremely short recording durations for these experiments. The table included in the manuscript shows that the recordings from these neurons are all less than 1 minute. The authors do not address this, arousing significant concerns. These elaborate experiments take a very long time to set up – why do the authors not record for longer time periods? This seems like a critical point. This is related to another issue: a compelling feature of the study is that the animals are awake (which is progress from slices and anesthetized animals), but they are not engaged in a task or any motor behavior (at least that is shown). This limits enthusiasm for the work. If the authors could record for longer, or even in different branches from the same cell, they could compare activity in running vs non-running, or examine the dendritic dynamics during a motor task. This might reveal exciting new data that would change how we think about dendritic integration and cerebellar function.

2018-03-30

NCOMMS-17-29817

Reviewers' comments:

Reviewer #1 (Remarks to the Author):

In this manuscript, Roome and Kuhn examine the integrative properties of Purkinje cell dendrites in awake mice. Using a technically awesome approach combining simultaneous dendritic Ca²⁺ and voltage imaging as well extracellular electrophysiological recording at the soma, they report several key details of Purkinje cell dendritic electrogenesis that are quite novel.

Thank you very much for your comments. We have addressed each of them and believe they have been very helpful in improving the quality of our analysis and in reinforcing our conclusions through the additional experiments we performed. We greatly appreciate your input.

Their observations include: 1) Spatio-temporal variance of climbing fiber evoked spiking and Ca²⁺ activity within a dendrite. 2) Dendritic spikes that occur independent of the soma through the putative activity of parallel fibers. 3) Subthreshold depolarizing potentials generated by putative parallel fiber activity that is correlated with simple spike output at the soma. These data extend current knowledge of Purkinje cell dendritic function and therefore should be of wide interest both to the cerebellar field as well as those interested in properties of dendritic integration.

Our primary approach in the additional experiments we have performed has been to replace the somatic recording electrode with a micropipette for drug delivery in awake mice (lines 101-109). Using pharmacology (Lidocaine and CNQX), we are now able to confirm that the subthreshold voltage fluctuations (and hotspots) we describe are due to excitatory synaptic input (parallel fibres). In addition, we have compared dendritic function in the awake mouse with that of the anaesthetized mouse (1% isoflurane) to examine the complex spike spatio-temporal variability, as several experts in the field have noted that this data would be very useful and highlights a key advantage of our technique (that the dendritic recordings were made in awake animals).

Enthusiasm for the work is limited by a number of concerns that should be straightforward to address.

1) Purkinje cell activity measurements were performed in resting mice (awake and but not behaving). Therefore, all methodological detail regarding monitoring behavior that isn't pertinent to the experiments performed should be removed or clarified (e.g., neural measurements were obtained during periods of inactivity as determined by...). Otherwise,

inclusion of such detail is confusing. For example, it is stated in the Methods that EEG recording was performed but there is no mention of EEG activity in the Results. Another example is whisker tracking measurements in Supplemental Figures 1 and 4 that are completely unrelated to the data presented nor are these measurements referred to in the text (as a side note, recordings were performed in an area of the cerebellum not known to be related to whisking).

We agree with this comment; we hope in future studies to include behaviour. For this study we have focussed on 'resting state' dendritic activity. The high variability and activity of dendritic function even in the 'resting state' is also very interesting. So for clarity we have removed the whisker movements and have updated the description in the methods. We include in the revised manuscript several experiments with isoflurane and therefore show an example of EEG recording to monitor the anaesthetized state of the mouse (i.e. to show sleep spindles) (Fig. S1).

2) The dendritic spike associated with the "startle" response is unconvincing. It occurs several hundred ms after the stimulus. Furthermore there is no reproducibility of this response. The "startle" DS spike is, apparently, only observed in one trial from one animal. Lastly, the authors present no evidence for a "startle" reflex (EEG?). It is advised that this data should be removed as it adds little to the manuscript. Perhaps this could be developed in future work.

We agree and have taken out the startle response. We will use it for future work.

3) The quantification and statistics that support claims regarding differences in "hot" and "cold" dendrites is not provided (i.e., more spikelets, higher spikelet peaks, spikelet spacing; lines 233-235). Relatedly, group data presented in Supplemental Figure 5 doesn't show a statistical difference in the average amplitudes of Ca²⁺ transients between "hot" and "cold" dendritic segments. Lastly, the authors conclude that variability in DCS signals occur due to spikelet generation and timing differences (lines 264 and 265) however there doesn't seem to be any quantification to support this claim.

We have now included the quantification and statistics for this, which have been included in the main text (lines 152-164) and figure 3. We also report significant modulation in DCS variability when awake and anaesthetized states are compared (lines 179-306).

4) The authors review possible factors that may contribute to spatio-temporal variability of the climbing fiber Ca²⁺ transient in the Results (Lines 268-276). This paragraph seems better

placed in the Discussion section because there are no follow-up experiments that require elaboration of these possibilities.

We agree, we have updated this section in light of our new experiments and findings (lines 171-306).

5) Do the putative parallel fiber evoked dendritic spikes show similar spatio-temporal variability in the dendrite as those evoked by climbing fibers? This should be quantified and reported for both the voltage and Ca²⁺ signals. That the DS spike appears widespread and not constrained to a branch or region is potentially very interesting. Does the widespread nature of the DS spike reflect propagation from a single initiation point in a specific branch due to the clustered activity of nearby parallel fiber inputs or does it result from the dendrite-wide activity of distributed but co-active inputs as inferred from the non-linear increase in percentage of hot spots shown in Figure 6g? This point should be clarified.

Yes, we agree and think the variability of DS events would be very interesting to follow up in a following study. As noted in your previous comment (2), and as we note in the text the DS are relatively rare. To study the spatial variability of this phenomenon in detail would require further experimentation.

6) Cross correlations between “slow” dendritic activity (hills and valleys) and simple spike rates (lines 342-343) aren’t shown.

We have now included this in Fig. 4. We also included the corresponding cross correlation between the calcium recording and SS rates as a control. These data confirm the relationship between the dendritic voltage recording and the SS rates and prove that the cross correlation is not an artefact of filtering (or cutting out DCSs, DS) because the corresponding calcium recording treated in the same way shows no correlation.

7) If the isolated, subthreshold voltage hotspots are AMPA EPSPS as speculated in the Discussion, aren’t they expected to decay passively along the dendrite? If so, how does their spatial domain (5-10 μm) compare to the expected length constant of the Purkinje cell dendrite?

We agree that this is a very interesting question but it will require detailed simulations which are beyond the scope of this manuscript. In general, we agree that the length constant is expected to be larger than 10 μm but we hypothesize that our size estimate for isolated hotspots is a result of morphological constraints, like the cross section of dendritic branchlets with spines (estimated diameter 5 μm) and a short branchlet length with many bifurcations in the distal dendrites.

8) Quantification of high spiking rates and percent of “hot” dendrite (lines 418-419) is not provided.

This quantification is shown in Fig. 6j.

9) In the legend of Figure 3, “d” is not included. It is probably mislabeled as “c”.

This has been corrected.

10) “DCS” is first used in line 129 but isn’t defined until lines 148-149.

We now define DCS earlier (Fig. 1. line 133 and line 143).

Reviewer #2 (Remarks to the Author):

In this technically oriented paper, the authors recorded 2P-excited fluorescence (1020 nm) of a linescan, scanned as superficially as possible but estimated at < 50 um below the pia mater, through the distal dendritic trees of 7 Purkinje Neurons. The neurons are expressing GCaMP6f and have been electroporated with ANNINE-6plus. An extracellular electrode is positioned close to the soma of the transfected/electroporated cell and an extracellular recording is made in parallel with the optical recording. The excited fluorescence is spectrally split and recorded on two detectors (490-550 nm and 550-750 nm), the interpretation being that one detector records ANNINE fluorescence and the other GCaMP fluorescence. The ANNINE trace is corrected for bleedthrough of the GCaMP fluorescence by subtracting off a scaled version of the GCaMP trace. The authors provide interesting hypotheses on the relation between dendritic and somatic activity in awake mice, but in general more work needs to be done before bolstered conclusions can be drawn.

Thank you very much for your interest and valuable comments. We have now increased our data set significantly, and using pharmacology are able to support our findings and make stronger conclusions, specifically with regard to your concerns on the impact of noise.

- The biggest player in dendritic heterogeneity, which is formed by the molecular layer interneurons providing inhibitory input to the Purkinje cells, is not taken into consideration. Please address at the experimental level.

This would be a very interesting study, but better accomplished using a different approach that allows access to single dendritic branches (as in Kitamura, Hausser 2011). The primary interest for our study is the effect that parallel fibres excitatory input has on dendritic heterogeneity, which until now could not have been addressed. Importantly, the most relevant previous study (Kitamura, Hausser 2011) already explored the impact of molecular interneurons on dendritic

heterogeneity in PCs in anaesthetised animals. The impact that anaesthetic has on cerebellar function is unclear. It is expected however that anaesthesia reduces excitatory input of PFs. We now compare the activity of the same PC under awake and anaesthetized states (Fig. 3g, h). Since our main experiments are performed on awake animals we tried to block GABA under awake conditions by local application of GABA blocker, but, unfortunately, had to abandon these experiments due to concerns about animal welfare.

- All records were focused on distal dendrites, as shown in Fig S2. The distal dendrites are unlikely to be innervated by climbing fibers; thus their responses indicate mainly the products of dendritic filtering rather than local CF inputs. Did the authors compare the responses from different parts of the Purkinje cell dendrites?

Yes, we did. These have now been included in Fig. S4. The CS responses we record look very similar at different locations as we mainly record from volumes with high surface to volume ratio, i.e. spiny dendrites. Unfortunately, we cannot easily record from aspiny dendrites as the noise level is too high.

- The authors claimed that the dendritic complex spike signals were variable. Using line scan in awake mice is prone to be affected by motion artifacts, especially the distortion in z axis. To what extent the dendritic complex spike variability is explained by instable recordings is unclear. In general, detailed analysis of recording stability is essential; please address this as well as the details highlighted below.

We have included further quantification of DCS variability to summarize the examples shown in Fig. 3. These show a clear relationship between the number of spikelets generated in different dendritic segments and the resulting calcium elevation. Motion artefacts that occur during the linescan recording invariably impact the focus of the entire dendrite and not individual segments, but this is not what we see. Consider the first single trial example shown in figure 3b and d for example (black arrows), which clearly show an additional spikelet restricted to a single dendritic segment with a corresponding time locked increase in calcium elevation. This could not be generated by a motion artefact. Our quantitative analysis also supports this observation and the fact that DCS variability actually increases in some PCs during anaesthetized states (during which motion artefacts would be minimized) further confirms our conclusions.

- The dataset appears to be somewhat over-interpreted at this point. This is exacerbated by the lack of a proper noise analysis. The 3 lines devoted to shot noise in the supplementary information makes me wonder what the authors mean by signal and by noise. An example of this is the claim in line 462: average amplitude of hotspot events is 6.8 ± 1.0 % DF/F mean \pm SD, shotnoise 2% DF/F. I may misinterpret the notation but this seems to say that the measured noise is smaller than the theoretically minimal noise.

This is a misinterpretation in this case. The estimate for hotspot amplitude ($6.8 \pm 1.0\%$) shows mean \pm SD between cells (i.e. we record a similar average hotspot amplitude across cells). The shot noise error associated with measuring the amplitude of a single hotspot is calculated to be 2%. We have rewritten this line for clarity (line 554). We have also revisited our analysis to include more stringent criteria for hotspot detection (included in supplementary methods) to further eliminate potential effects of noise. The most conclusive approach however, to test if the voltage fluctuations are due to noise (biological or systematic) or as we propose, due to synaptic activity, is to use pharmacology. We have now conducted many more experiments using pharmacological blockers (Lidocaine and CNQX) and by comparing awake state with anaesthetized state we can confirm that the subthreshold voltage fluctuations and hotspots that we detect are not due to noise.

- The measurement is a linear line-scan through a set of dendrites. Interpretations of the effect of different timing in different part of the scan appears to be done without any correction/discussion of the actual difference in propagation length/time from the recorded points to branching points in the dendrite or to the soma, which raises doubts on the concept of relative timing of the signals in the different parts of the scan.

The sequential data acquisition during a line scan will affect analysis of signals in the sub millisecond range. The temporal variability that we see in DCS signals (Fig. 3 and S6) is far greater than the 2kHz sampling frequency that we use.

- Figure 2b: variability in spikelet amplitude seems to follow a normal noise distribution as expected for an optical recording with limited photons. The interpretation in the text is however that there is an underlying physiological phenomenon (line 170-171). Without proper noise analysis this conclusion cannot be supported.

See comment above on noise analysis.

- In lines 172 to 179 an interpretation of the data is provided that emphasizes the superiority of the optical recording over previous efforts, but it is noted that an expected secondary plateau potential is missing. This could be a signal to noise issue, but it could also mean that the recorded traces do not accurately reflect voltage waveforms. Without a proper noise analysis and most importantly without any direct measurement comparing a known voltage signal to the optical recording, it is not possible to interpret the data in a meaningful way.

This is a misinterpretation. We do not claim a superiority of the optical recording over previous efforts. We claim that the optical data has a different origin (thin spiny distal dendrites with a large surface area) than the previous data (thicker aspiny dendrites). Therefore, the optical method is a complementary tool and our findings are an addition to the existing data. We adapted the manuscript to clarify (lines 185-188).

- This reviewer wonders to what extent the authors can distinguish voltage-modulated fluorescence from non-voltage-modulated fluorescence. I wonder whether they can interpret a DF/F in one mouse as a spikelet amplitude and compare it to DF/F and spikelet amplitudes in another mouse, as levels of background fluorescence/tissue auto-fluorescence and non-voltage sensitive fluorescence are going to be different. Yet this is exactly what is done, which makes all conclusions drawn about relative magnitudes of spikelets suspect. A background correction is applied, but this can make things worse. For example, in figure S5 the difference in amplitude between spikelets of hot and cold segments is emphasized. The voltage trace is obtained by subtracting a scaled version of the GCaMP trace. The undershoot after the spike in the cold segment makes it look as if simply a slightly wrongly scaled version of the calcium trace was subtracted, rendering the meaning of the resulting difference in spikelet amplitude uncertain. Given the small difference in spikelet amplitude and the nonlinearity in voltage response of ANNINE-6plus, it is also hard to say whether spikelet sizes of different mice can be compared, as no measure of baseline voltage is available.

See comment above on noise analysis. We show the DCS amplitude recorded from all PCs in Fig. 2b. Here we found that even though the DCS are highly variable in amplitude for a given cell, their average amplitude is remarkably similar between cells, this could justify comparing DCS responses between cells. But in general this is not what we do. Our aim is to investigate the variability within the same cell and even within the same DCS signal, which should be assumed to have the same levels of background fluorescence/tissue auto-fluorescence and non-voltage sensitive fluorescence (and also GCaMP6f fluorescence) within a single cell, especially considering the long (24 hours) diffusion time following ANNINE-6plus injection.

Following the argument on GCaMP trace scaling through to completion, this would indicate that the largest calcium transients would cause the strongest undershoot effect. If anything, this would reduce the number of detected spikelets, but instead we see the opposite – an increased spikelet count corresponding to the largest calcium transient (now shown in figure3), any errors in the calcium trace scaling would have a confounding effect on what we see.

- The authors give Kuhn 2004 as a reference though this deals with ANNINE6 rather than ANNINE-6plus; given the similarity in Fromherz 2008 of the spectra, and the fact that the authors base voltage sensitivity on Kuhn 2004, I assume the same nonlinearity holds too, but please address the issue in the discussion.

ANNINE-6 and ANNINE-6plus share the same chromophore and the same mechanism of voltage sensitivity. Only the solubility is different due to a different head group. The spectral shift in both chromophores is linear which results in a slight nonlinearity of the voltage response for very large membrane voltage changes. For voltage changes relevant for this paper (less than 50 mV voltage change) the nonlinearity can be neglected (lines 75-76).

In general, the development and the details of the methods which we applied in this paper is a long story and we agree that it is necessary to read in detail several of our previous publications to understand in depth some of the methodological details. Therefore, we recently wrote a book chapter summarizing and explaining all the details up to the present paper (excluding any data of the current manuscript). The book (Nature Protocols, Multi-Photon Microscopy) will go in print in summer 2018. We will upload the current draft of this book chapter for your reference.

- In figure 4 an average SS is calculated and convoluted with the SS binary trace. Much is made of the fact that this trace shows less variation in DF/F than the actual trace of dendritic voltage. The averaging has effectively filtered out one source of noise in that shape, i.e. the spike to spike variation. It is therefore not surprising that figure 4d and e look like they do. Other traces of random noise treated the same way may show similar behavior. Please address.

We have now included pharmacology experiments in Fig. 4e and f supporting our conclusion and proving that the variability in the subthreshold voltage signals are due to excitatory parallel fibre synaptic input.

- In figure 1f and g, their spatiotemporal maps are filtered with a 5 μm boxcar spatial filter and a 5 ms temporal filter. Effectively this means that they re-sampled their data to 5 μm and 5 ms (disregarding some minor effects of the actual filter shape), limiting their detection bandwidths to 0.2/ μm and 200 Hz. They then interpolate the data again at 10 kHz, but this does not change the fact that they filtered all higher frequency information out of their signal. This may invalidate their hotspot analysis. They select hotspots > 3 ms and 1.5 μm , which effectively means single pixel noise at these bandwidths. This means that for instance the average temporal size of isolated hotspots (5.5 +/- 0.8 ms) and average spatial width (4.4 +/- 0.7 μm) are hard to interpret, as they're looking at their detection bandwidth. The meaning of Figure 6a is therefore uncertain. The authors aim to show that crosstalk in the green channel shows a voltage signal, but that spike only means that the noise is not detector noise and does not exclude any common mode noise, i.e. in the laser intensity or detection electronics or even biological noise. Averaging random noise peaks, centered specifically at the maximum amplitudes of said noisepeaks, may give the same effect that the authors claim proves the voltage nature of their signal.

Thank you for identifying this unclear description of our analysis. We use the filtered spatio-temporal maps to detect hotspots, but for determining the hotspot profiles we use the original unfiltered data (we now added lines 517-518 to clarify). We agree that hot pixels could contribute to hot spots. Therefore, we introduced some more stringent selection criteria for hotspot detection. We also are able to reduce hotspot size and frequency significantly with lidocaine and CNQX. See Fig5 e-i.

Also, we repeatedly tested our setup to make sure that it is shot noise limited. If this was not the case, artefacts were obvious immediately as our recordings are extremely noise sensitive. For example, laser noise is not correlated with line scanning and results therefore in bright and

dim lines; the same is true for most other electric noise sources resulting in regular patterns in the spatio-temporal map; movement artefacts distort the lines. In contrast, hotspots occur in consecutive lines (temporal dimension) at the correct spatial position.

- The meaning of Figure 6h is unclear. The authors filtered out the DCS and CS events from their hotspot trace, which are also missing from the SS binary. A convolution of those traces would, based only on these artificially created correlated events alone, already show a peak around 0 lag, which would disappear with time reversal of one of the traces.

This is an important point. We repeated this analysis in Fig. 6h, this time by sectioning the data into 500ms segments and selecting only data that do not contain DCS or DS for the cross correlation analysis. Here it was no longer necessary to remove (or filter out) DCS and DS events, yet there remained a clear correlation between the dendritic hotspot events and somatic SS activity, that disappeared when the traces are reversed.

- The authors base a lot of their discussion and interpretation on the spatial range of their linescan and the average number of dendrites and spines in the thus defined imaged lines that they expect from literature. I would argue that, for a paper that purports to spatially and temporally disentangle voltage dynamics in these subcellular regions, this is not a best practice.

We agree that it would be better to image a neuron first in vivo and then to do an EM reconstruction. We plan to do this in the future but it is beyond the scope of the current paper.

- Linelengths are defined as $\sim 200 \mu\text{m}$ or $256 \mu\text{m}$; at 512 pixel lines, this changes the number of pixels in a $5 \mu\text{m}$ boxcar filter, or the number of pixels in the lengthscale that is claimed as significant for hotspots. They further state that they have done a large part of their analysis on ROI, where they cropped out areas without dendritic membrane. This last action would presumably throw off the average number of dendritic spines in a volume, changing the discussion. It is not clear with relation to what space these average numbers have been defined. They further claim their focal spot is $1 \mu\text{m}$ and show an image of their focus that was made by scanning a $1 \mu\text{m}$ bead through the laser focus. This gives an imaged spot that, according to their scale bar, is also $1 \mu\text{m}$. A $1 \mu\text{m}$ focus convoluted with a $1 \mu\text{m}$ bead would however not give a $1 \mu\text{m}$ spot, so either their focus is of a significantly different size than they think, throwing off their estimates for spine and dendrite numbers in the focal volume, or the spatial range of the experiment is off. Please address this issue.

All linescans were recorded with $256 \mu\text{m}$ length and 512 pixels, so that 1 pixel corresponds to $0.5 \mu\text{m}$. So, the filtering is always the same. All of our ROIs for hotspot analysis was done on 100 pixels ($50 \mu\text{m}$ dendritic segments) of the 512pixel linescan, without any cropping within the 100pixel segment (see Fig. S8).

The 1 μm bead was chosen to resemble a typical dendritic structure (e.g. dendritic shaft diameter). So, the image does not show the point spread function. We checked the scale bar again. The image of the 1 μm bead is slightly widened but significantly elongated in z direction. The red halo around the yellow and white center is due to the convolution (Fig S10).

Reviewer #3 (Remarks to the Author):

The manuscript by Roome and Kuhn leverages an impressive combination of challenging neurophysiological techniques to assess synaptic integration in cerebellar Purkinje neurons of awake mice. Despite decades of investigation into the mechanisms of dendritic function, few studies have been able to address how dendrites influence input processing in vivo, particularly in awake animals. Those few studies have generally employed Ca^{2+} imaging, which is an indirect report of dendritic voltage. Here, the authors combine Ca^{2+} imaging with voltage-sensitive dye imaging to quantify the organization of subcellular voltage changes in distal dendritic branches as a function of somatic output. The methodological approach is remarkable: the authors label single neurons with voltage sensitive dye via two-photon guided electroporation after an earlier surgery to infect many neurons with GCaMP; they allow the dye to spread, and then reintroduce a pipette to record juxtacellular signals from the neuron with the VSD. The ability to index dendritic signals in both the voltage and Ca^{2+} domains by the somatic output pattern is tremendously useful (and impressive). While I have two experimental concerns (listed below), a major issue with the manuscript is in regards to the impact of the presented data. Its unclear what the findings of the study are, and what they mean for the field. The work is highly descriptive; as a result, I cannot quite figure out how this should change our thinking for how cerebellar dendritic integration works. The authors do not frame their results in the context of testing a hypothesis, or disambiguating prior controversies, or uncovering novel features of Purkinje neuron function. In fact, they mention that what they see in vivo matches what has previously been described in slices during direct dendritic recording. As it stands, it is hard to know what the manuscript is trying to say, aside from demonstrating that this kind of recording is possible.

Therefore, its significance for the field is limited.

We thank Reviewer #3 for acknowledging the experimental challenges of our work.

Regarding the significance of our work we argue in the following way:

The manuscript under review is partly a methods paper. We reflect this in the title. However, we think it is more than demonstrating that this kind of recording is possible.

The manuscript is a necessary step to validate our methodological advances as voltage imaging is still a very challenging technique. For example: We find that labelled Purkinje Neurons

survive many days and even weeks in vivo and that functional signals can be repeatedly recorded during this time (new Fig S3). The simple and complex spike rates which we recorded at the soma matches previously published in vivo electrophysiological recordings. Our imaging is optimized for low phototoxicity and high voltage sensitivity (Kuhn, Fromherz, Denk 2004). In our recordings we do not correct for bleaching which is associated with photodamage. We can record not only 50s but 500s from a single dendrite.

Our work confirms many of the findings from in vitro work. We think this confirmation is an important step to really trust the previously collected data from brain slices. Therefore, our data which confirms previous findings does not change how we think about the cerebellum but it confirms that the assumptions on which our ideas are based are facts and hold in the awake animal. We think that this is an important step forward.

The manuscript also includes several new findings. For example: CF-triggered complex spikes have a highly heterogeneous effect on spatial patterns of voltage, including failures. Therefore, CF activity is not at all the monolithic event that people suppose. Spiny PC dendrites have rich local signalling, consistent with clustered PF input. Local signals can summate to make local regenerative events.

We agree that our findings were so far very descriptive. This is due to the challenges of doing the experiments which go far beyond all existing experiments (for the first time simultaneous voltage and calcium imaging; for the first time simultaneous voltage and calcium imaging with electrical recordings; for the first time all this in an alive animal; and for the first time all of this in an awake animal). That the paper became more descriptive than we would wish is also due to the overwhelming richness of information in our multidimensional data.

However, during the past 3 months we added several pharmacological experiments to add a mechanistic understanding. We show how lidocaine and isoflurane affects the signalling of complex spikes and show how hotspots are affected by different drugs.

In the first manuscript we reported data from 7 PNs. Now we increased it to more than 30 PN and added – another novelty - pharmacology experiments with simultaneous voltage and calcium imaging from dendrites in awake animals.

Regarding the technical approach, the use of voltage dyes often presents serious experimental caveats both in terms of altering the properties of the recorded neurons and interpreting the resulting signals. The authors cite previous studies of this particular dye, but there is very limited physiological quantification in these papers. Given how much work the authors invest in their labor and time intensive in vivo experiments, a separate set of slice experiments to rigorously evaluate the dye seems necessary to me. Otherwise, it is unclear what the VSD signals recorded in vivo represent. Ideally, the authors would measure the linearity of the dye across a range of voltage amplitudes and kinetics from a structure in which they have a ground-truth patch-clamp electrical recording (i.e. the soma or a proximal dendrite), as well as

critically evaluate any potential disturbances in the neuron's membrane properties produced by loading with the dye.

The voltage-sensitive dye used in this study, ANNINE-6plus, is arguably the best studied existing voltage-sensitive dye. It has an optimized design based on a number of experimental and theoretical studies:

- Ephardt H, Fromherz P (1991) Anilino-pyridinium - Solvent-dependent fluorescence by intramolecular charge-transfer. *J Phys Chem* 95:6792-6797
- Ephardt H, Fromherz P (1993) Fluorescence of amphiphilic hemicyanine dyes without free double-bonds. *J Phys Chem* 97:4540-4547
- Röcker C, Heilemann A, Fromherz P (1996) Time-resolved fluorescence of a hemicyanine dye: Dynamics of rotamerism and resolution. *J Phys Chem* 100:12172-12177
- Fromherz P (1995) Monopole-dipole model for symmetrical solvatochromism of hemicyanine dyes. *J Phys Chem* 99:7188-7192
- Fromherz P, Heilemann A (1992) Twisted internal charge-transfer in (aminophenyl)pyridinium. *J Phys Chem* 96:6864-6866
- Lambacher A, Fromherz P (2001) Orientation of hemicyanine dye in lipid membrane measured by fluorescence interferometry on a silicon chip. *J Phys Chem B* 105:343-346
- Hübener G, Lambacher A, Fromherz P (2003) Anellated hemicyanine dyes with large symmetrical solvatochromism of absorption and fluorescence. *J Phys Chem B* 107:7896-7902

Most importantly, we were able to show that ANNINE-6 and ANNINE-6plus (same chromophore but less hydrophobic than ANNINE-6) are pure electrochromic probes:

- Kuhn B, Fromherz P (2003) Anellated hemicyanine dyes in a neuron membrane: Molecular Stark effect and optical voltage recording. *J Phys Chem B* 107:7903-7913
- Fromherz P, Hübener G, Kuhn B, Hinner MJ (2008) ANNINE-6plus, a voltage-sensitive dye with good solubility, strong membrane binding and high sensitivity. *Eur Biophys J Biophys* 37:509-514

A pure electrochromic probe is independent of the differences of membrane composition between different species and cell types. Also, spectral shift is linear for depolarizations and hyperpolarizations. The resulting fluorescence change is slightly nonlinear for large voltage changes but can be considered linear for voltage changes below 50 mV as in this study. So, ANNINE-6 and ANNINE-6plus work in all biological cells. The temporal resolution is 8 ns.

To use ANNINE-6 dyes optimally we developed a novel imaging method:

- Kuhn B, Fromherz P, Denk W (2004) High sensitivity of Stark-shift voltage-sensing dyes by one- or two-photon excitation near the red spectral edge. *Biophys J* 87:631-639

This method increases the sensitivity and reduces phototoxicity dramatically. This method is based again on the pure electrochromic behavior of ANNINE-6. We find the same behavior of

the dye in leech neurons and HEK293 cells. If the dye would behave differently in mouse PNs than in leech neurons or HEK293 cells this method would not work and we would not be able to get any voltage signal.

In the first version of the manuscript we report 50 seconds of recordings. However, the limiting factor is not the imaging but mainly the electrical recording in awake mice. In pharmacology experiments which we added in response to the reviews we record for more than 500 seconds without detrimental effects. These simultaneous dendritic voltage and calcium recordings include control, drug application, and washout. We did not observe bleaching which is associated with phototoxicity. If anything would be different from our previous experiments in leech neurons and HEK293 cells (let's say a spectral shift or a different mechanism) the experiments would not work.

We also show that we can repeatedly record from PNs for at least 2 weeks (Fig. S3).

We also added recordings from different depth of the dendritic tree (Fig. S4). Recordings from the soma are difficult because simple spikes in vivo last for less than 500 μ s which is less than our sampling rate.

In the manuscript under review many developments came together. To summarize and describe them is beyond the scope of the paper. Therefore, we wrote a book chapter which summarizes all developments so far. We submit the book chapter with our response to the reviews for your reference.

A further experimental concern pertains to the extremely short recording durations for these experiments. The table included in the manuscript shows that the recordings from these neurons are all less than 1 minute. The authors do not address this, arousing significant concerns. These elaborate experiments take a very long time to set up – why do the authors not record for longer time periods? This seems like a critical point. This is related to another issue: a compelling feature of the study is that the animals are awake (which is progress from slices and anesthetized animals), but they are not engaged in a task or any motor behavior (at least that is shown). This limits enthusiasm for the work. If the authors could record for longer, or even in different branches from the same cell, they could compare activity in running vs non-running, or examine the dendritic dynamics during a motor task. This might reveal exciting new data that would change how we think about dendritic integration and cerebellar function.

The short recording time of about 50 seconds for each PN is due to the limited time of electrical recordings under awake conditions. Now we added data without electrical recording where we imaged for more than 500 seconds without deteriorating effects or bleaching. We also show that we can record for many days.

It is also important to notice that we image at 2 kHz from the same dendrite. If we record for 500 seconds we sampled the dendrite $2000\text{Hz} * 500\text{s} = 1,000,000$ -times. For calcium imaging a

typical sampling rate is 30 Hz. To sample 1,000,000 times at this rate it takes 9 hours. We thank the reviewer for his/her suggestion to record during different behavior. We are currently working on these experiments and we have very promising preliminary results. However, we think this is beyond the scope of the here submitted manuscript.

Reviewers' comments:

Reviewer #1 (Remarks to the Author):

The authors have added significant new data that bolster their claims and help alleviate my concerns. It is understandable that exploration of spatio-temporal variability of DS spikes (point 5) is hampered by the rarity of their occurrence and therefore is not pursued in the revision. Regarding passive spread of hotspots (point 7), the authors should consider adding a sentence in the discussion highlighting the abbreviated spatial extent of hotspots and the potential limits of their approach as detailed in their reviewer response. This would be helpful for readers and in no way would detract from the quality of the work presented and the validity of their conclusions.

Reviewer #2 (Remarks to the Author):

I congratulate the authors on a much improved manuscript. I have no further concerns.

Reviewer #3 (Remarks to the Author):

In this revision, Roome and Kuhn have increased their sampling to include more neurons (some with longer optical recording durations), performed new pharmacological experiments, and included further analyses to address the many comments they received from reviewers. However, the revised manuscript and its accompanying rebuttal do not satisfyingly address the core criticisms raised in the previous round of review.

1. While I can appreciate that this manuscript is “partly a methods paper”, the authors appear to be trying to have their cake and eat it too. Methods papers should present a breakthrough technology or approach, rigorously defined and delineated in its technical detail, with a strong focus on its potential utility for the larger scientific community. Conventional scientific reports must present new discoveries, test hypotheses, or resolve controversies. This manuscript attempts to take a middle path, but currently lacks both an acceptable level of technical rigor as well as significant scientific impact.

There are novel features of the author’s approach, as they point out (“for the first time simultaneous voltage and calcium imaging; for the first time simultaneous voltage and calcium imaging with electrical recordings; for the first time all this in an alive animal; and for the first time all of this in an awake animal, another novelty- pharmacology experiments with simultaneous voltage and calcium imaging from dendrites in awake animals”). But just because it is novel does not make it automatically meaningful or useful to the community. It must be technically robust and must lead to some important biological insight.

2. The entire manuscript is based on optical recordings of a voltage dye which, as far as has been presented by the authors, has not been tested for use in mammalian neurons. In the

previous round of review, I expressed that ground truth validation of this dye in brain slices using patch clamp electrophysiology was necessary. The authors have chosen to ignore this absolutely crucial control experiment. The authors clearly feel very strongly that this dye works well (otherwise why do so many challenging experiments with it); they should want to rigorously describe its performance and demonstrate directly, to themselves and to others, what the signals they record mean in terms of dendritic voltage. Otherwise, we will all be left wondering how much of the data and conclusions obtained with this dye are robust.

As far as I am aware, all other voltage dyes and genetically encoded voltage sensors that are used for somatic or dendrite imaging are required to have been tested with patch clamp physiology in neurons (either in slices or cultures). Without this critical control, the data are impossible to interpret.

Unfortunately, the authors cite many irrelevant papers in their rebuttal to this point (all of which are from Fromherz). This comes off like an attempt to obfuscate the issue. The exact chemical/optical properties of the dye in solution (or in HEK cells or in leech cells) are irrelevant. The authors must test if the dye alters the electrical properties of PN neuronal membranes (e.g. by increasing C_m) or ion channels (as some other voltage dyes do) by independently measuring the electrical properties with patch-clamp physiology. If this is indeed the case, any *in vivo* data obtained with the dye will be contaminated. The degree of contamination must at the very least be measured and could potentially be compensated for in a principled way. Additionally, despite numerous claims in the field that previous voltage dyes behave linearly, this is not always the case and requires careful testing using the actual calibrated voltage signals one is trying to measure.

3. If the authors do indeed demonstrate that the dye works as advertised, concerns remain about the significance of the scientific conclusions of this study. The authors have: 1 - Potentially "confirmed" some previous *in vitro* findings in awake animals. This is an insufficient advance in my view. And 2 - New pharmacological studies show that distal PN dendrites in awake animals have local spatio-temporal voltage fluctuations that are dependent on synaptic input. Of course they do. How could it work otherwise? This is only interesting if the authors were to observe the converse; that somehow dendritic spatio-temporal voltage is not dependent on synaptic input.

4. Why are the authors only able to record electrically from their neurons for less than 1 min? Many groups routinely perform whole-cell patch-clamp recordings (which are much more difficult and more sensitive to movement than the cell-attached recordings used by the authors) during behavior in mice.

5. Number of samples is not the issue – the length of time across which the sampling is conducted is what matters in this particular case. There is no compelling reason to sample GCaMP6 much more than 30 Hz based on the kinetics of the dye and the underlying Ca^{2+} signal. Of course, voltage is much faster and it must be sampled as such. What is the

rationale for recording from dendritic segments for 500 s? Whatever it is, surely this should be enough to correlate dendritic signals with behavioral features (running, whisking, pupil dilation, posture, etc). Without any sort of behavioral result, it is still unclear what any of the recorded signals have to do with anything.

I appreciate that the authors feel that behavior is beyond the scope of the current manuscript, but I disagree, particularly in conjunction with the stated technical concerns. The methods component of the manuscript is not strong enough to carry a purely descriptive scientific result. Substantial, novel conclusions regarding general principles for dendritic integration and/or PN/cerebellar function in vivo during behavior would greatly strengthen this study.

In summary, the authors' approach has significant promise, but ground truth validation is necessary, as is correlation of dendritic signals with behavioral variables.

Reviewers' comments:

Reviewer #1 (Remarks to the Author):

The authors have added significant new data that bolster their claims and help alleviate my concerns. It is understandable that exploration of spatio-temporal variability of DS spikes (point 5) is hampered by the rarity of their occurrence and therefore is not pursued in the revision. Regarding passive spread of hotspots (point 7), the authors should consider adding a sentence in the discussion highlighting the abbreviated spatial extent of hotspots and the potential limits of their approach as detailed in their reviewer response. This would be helpful for readers and in no way would detract from the quality of the work presented and the validity of their conclusions.

We are happy to adapt the manuscript as requested. See line 627 – 634 in the manuscript. Thank you very much for your valuable comments!

Reviewer #2 (Remarks to the Author):

I congratulate the authors on a much improved manuscript. I have no further concerns.

Thank you very much for your valuable comments!

Reviewer #3 (Remarks to the Author):

In this revision, Roome and Kuhn have increased their sampling to include more neurons (some with longer optical recording durations), performed new pharmacological experiments, and included further analyses to address the many comments they received from reviewers. However, the revised manuscript and its accompanying rebuttal do not satisfyingly address the core criticisms raised in the previous round of review.

1. While I can appreciate that this manuscript is “partly a methods paper”, the authors appear to be trying to have their cake and eat it too. Methods papers should present a breakthrough technology or approach, rigorously defined and delineated in its technical detail, with a strong focus on its potential utility for the larger scientific community. Conventional scientific reports must present new discoveries, test hypotheses, or resolve controversies. This manuscript attempts to take a middle path, but currently lacks both an acceptable level of technical rigor as well as significant scientific impact.

There are novel features of the author’s approach, as they point out (“for the first time

simultaneous voltage and calcium imaging; for the first time simultaneous voltage and calcium imaging with electrical recordings; for the first time all this in an alive animal; and for the first time all of this in an awake animal, another novelty- pharmacology experiments with simultaneous voltage and calcium imaging from dendrites in awake animals”). But just because it is novel does not make it automatically meaningful or useful to the community. It must be technically robust and must lead to some important biological insight.

We disagree, as do Reviewers #1 and #2. Reviewer #3 lists several of our technical advances. We list our main physiological findings now very clearly shown in the abstract:

- Discrete 1-2 ms supra-threshold voltage spikelets were detected during dendritic complex spikes in the distal spiny dendrites. Spikelets were highly heterogeneous in number, timing and spatial distribution within single complex spikes and between complex spikes. This heterogeneity was reflected by spatio-temporally variable dendritic calcium signals.
- Sub-threshold voltage imaging detected highly attenuated back-propagating action potentials, and spatio-temporal voltage maps revealed highly variable 5-10 ms sub-threshold ‘hotspots’ localized to fine dendritic processes, which were reduced in size and frequency by lidocaine and CNQX.
- ”Hotspots” correlated with somatic output but also, at high frequency, triggered purely dendritic calcium spikes.

These findings would be not possible without the technical advances. Thereby we show to the community that it is meaningful and useful.

The aim of our project was to collect more information from a single neuron in vivo than had been done previously. Our data mutually supports and confirms earlier studies, that were limited to slice experiments or through theoretical techniques. This does not diminish the significance of our findings. For instance, simply because non-climbing fiber evoked dendritic spikes and dendritic hotspots CAN occur in slices or in computational models, doesn’t mean that they DO occur in vivo. In our study we now prove that they occur and we describe how they occur in the resting state of an awake animal.

Moreover, using simultaneous dendritic voltage imaging, dendritic calcium imaging, and somatic extracellular recording, we reveal several novel aspects of neuronal function in vivo that could not be predicted by slice experiments or by theoretical models, such as the non-linear relationships between (physiological) dendritic hotspot activity and SS output. This is because prior to our study the dendritic hotspot activity remained unexplored experimentally.

2. The entire manuscript is based on optical recordings of a voltage dye which, as far as has been presented by the authors, has not been tested for use in mammalian neurons. In the previous round of review, I expressed that ground truth validation of this dye in brain slices using patch clamp electrophysiology was necessary. The authors have chosen to ignore this absolutely

crucial control experiment. The authors clearly feel very strongly that this dye works well (otherwise why do so many challenging experiments with it); they should want to rigorously describe its performance and demonstrate directly, to themselves and to others, what the signals they record mean in terms of dendritic voltage. Otherwise, we will all be left wondering how much of the data and conclusions obtained with this dye are robust.

As far as I am aware, all other voltage dyes and genetically encoded voltage sensors that are used for somatic or dendrite imaging are required to have been tested with patch clamp physiology in neurons (either in slices or cultures). Without this critical control, the data are impossible to interpret.

Unfortunately, the authors cite many irrelevant papers in their rebuttal to this point (all of which are from Fromherz). This comes off like an attempt to obfuscate the issue. The exact chemical/optical properties of the dye in solution (or in HEK cells or in leech cells) are irrelevant. The authors must test if the dye alters the electrical properties of PN neuronal membranes (e.g. by increasing C_m) or ion channels (as some other voltage dyes do) by independently measuring the electrical properties with patch-clamp physiology. If this is indeed the case, any *in vivo* data obtained with the dye will be contaminated. The degree of contamination must at the very least be measured and could potentially be compensated for in a principled way. Additionally, despite numerous claims in the field that previous voltage dyes behave linearly, this is not always the case and requires careful testing using the actual calibrated voltage signals one is trying to measure.

We believe this comment is largely due to a misunderstanding. In short, calibration experiments using ANNINE-6 and ANNINE-6plus have been performed and are published already, both *in vivo* (Kuhn, Denk, Bruno, PNAS, 2007, '*In vivo two-photon voltage-sensitive dye imaging reveals top-down control of cortical layers 1 and 2 during wakefulness*') and on cultured neurons by two independent groups (Pages et al, 2011, Front Cell Neurosci, '*Optophysiological approach to resolve neuronal action potentials with high spatial and temporal resolution in cultured neurons*') and (Mennerick, Chisari et al. 2010, J Neurosci, '*Diverse voltage-sensitive dyes modulate GABAA receptor function*'). These studies provide experimental controls that prove the reliability of ANNINE-6plus for voltage imaging in neurons.

Specifically, in two comprehensive studies, ANNINE-6plus was tested using patch clamp recording on cultured neurons (Pages et al. 2011) and (Mennerick et al. 2010). These studies from independent groups proved that ANNINE-6plus is a reliable voltage sensor for neurons. The authors even claim by comparing several voltage sensitive dyes that ANNINE-6plus is more reliable than more commonly used voltage sensitive dyes for studying neuronal function. For example, the authors state that '*ANNINE6 and ANNINE6plus had no detectable effect on GABAA receptor function.*' (Mennerick et al. 2010), whereas the other voltage sensitive dyes they tested had modulatory effects.

We admit that these points were not made clear enough in our initial response. To be clear; the purpose of our study was not to introduce and characterize a novel voltage sensitive dye. As

Reviewer #3 correctly points out “As far as I am aware, all other voltage dyes and genetically encoded voltage sensors that are used for somatic or dendrite imaging are required to have been tested with patch clamp physiology in neurons (either in slices or cultures). Without this critical control, the data are impossible to interpret”. We agree with this and note that the same critical controls have also been carried out for ANNINE-6plus and since its development, ANNINE-6 and ANNINE-6plus, have been successfully used to study a broad range of cell types, including neurons in vitro and in vivo, (Kuhn, Denk, Bruno, 2007) (Bu et al. 2009) (Flickinger et al. 2010) (Vignali et al. 2010) (Mennerick et al. 2010) (White et al. 2011) (Pages et al, 2011) (Berghoefer et al. 2012) (Ramamoorthy et al. 2013) (Wegner et al. 2013) (Silve et al. 2015).

In our initial response to Reviewer #3’s comments, we did not perform the requested slice experiments in Purkinje neurons because they are both technically impossible and unlikely to yield a meaningful outcome. We would like to clarify our reasons for this and the alternative approaches and experimental controls that we provide. We also would like to explain why the previously published studies performed by independent groups characterizing ANNINE-6plus, are sufficient to support our data.

We agree that an additional cerebellar slice experiment appears to be a relatively simple and useful supplement to our study, but we did not perform this experiment due to two critical problems, that we would like to outline below:

Problem 1: The experiment Reviewer #3 requests is technically impossible.

Unlike other voltage sensitive dyes, ANNINE-6plus has never been successfully applied to in-vitro slice experiments (although we and other groups have tried!). This is due to technical difficulties. Reviewer #3 asks to test if the dye responds linearly in a mammalian neuron, specifically in Purkinje cells in slices. Equivalent control experiments have been performed on labelled cultured cortical neurons with ANNINE-6plus (labelled by bath application). These calibration experiments proved that the dye behaves linearly within the physiological range, and did not alter the electrical properties of the neuron, in the soma and dendrites (Pages et al. 2011).

But this experiment has not yet been performed on neurons in slices. The most common approach for labelling neurons with voltage sensitive dyes in slices is via a patch pipette (Antic & Zecevic, 1995) (Palmer et al. 2010). Unfortunately, ANNINE-6plus is too insoluble to be dissolved in patch pipette solution and will typically block the pipette tip to prevent labelling. To quote the authors of the publication supporting this claim: ‘We first tried [labelling] via patch clamping, with the dye in the patch pipette. Surprisingly, the dye did not penetrate into the cell, even though other dyes, such as Alexa594 filled the neurons normally. We reasoned that the solubility of the dye in the intracellular solution was not sufficient, leading to small crystals that plugged the patch pipette’ (Pages et al. 2011).

The method that we developed for in vivo labelling uses electroporation instead, after which it is necessary to allow >12hrs for the dye to diffuse throughout the cell, and also away from the extracellular space. Dye within the extracellular space following electroporation will produce background fluorescence, but will not contribute to the voltage signal and will thereby dramatically dilute the voltage signal. This is especially problematic for Purkinje cells because of

basket cell labelling which densely ensheathes the Purkinje soma. Thus, this labelling approach is simply not possible in slices.

We attempted to first label Purkinje neurons in vivo, to allow for dye diffusion before slicing the cerebellum. We were unsuccessful in locating the labelled neurons in vitro however, and concluded that performing control slice experiments in this way was extremely difficult and low yield, and still unlikely to yield meaningful results (see Problem 2).

Problem 2: The experiment Reviewer #3 requests is unlikely to yield meaningful results.

All our voltage imaging experiments were performed in the most distal dendrites and we even argue that most of the signal comes from spines. This being so, the calibration experiment performed at the soma (or even primary or secondary dendrite) would not be sufficient for calibrating the voltage signals in the distal dendrites. This is because a voltage-clamp recording from a neuronal soma (or even dendrite) by patch clamp electrophysiology cannot faithfully provide a voltage measurement of the distal dendrites, this ‘space-clamp’ limitation of the patch clamp technique is well known. Thus the only adequate control experiment would be to patch the very distal dendrites and simultaneously do voltage imaging at that location. However, this experiment is also impossible.

Therefore, the most reliable ‘ground truth’ calibration experiment is to simultaneously patch clamp and image voltage from the same location. Incidentally, this type of experiment is most accurate in HEK293 cells (Kuhn et al. 2004) and cultured leech neurons (Fromherz et al. 2008) or alternatively at the soma of cultured neurons (Figure below) (Pages et al, 2011), where the voltage recorded by the patch clamp electrode is closest to iso-potential throughout the dye labelled membrane at that location. We regret not making these technical points from the outset.

Optical recordings with ANNINE-6plus of action potentials and low amplitude depolarizations. (A) Top Inset a typical line scanned along the soma is illustrated in red. Scale bar 10 μm . Top: The stimulation protocol consists in a ramp of 14 steps (30 ms) ranging from -40 to $+100$ mV imposed in voltage clamp mode. Middle: Fluorescence detected in line scan mode at the soma synchronously with stimulation protocol. Bottom: From this image, the ANNINE-6plus fluorescent time profile is extracted. (B) ANNINE-6plus fluorescence intensity changes upon amplitude of membrane depolarization. (C) Low amplitude depolarizations (voltage ramps, 5 steps/10 ms, ranging from -60 to -50 mV) optically detected at the soma by averaging 30 (top trace) and 100 (middle trace) single trials. (D) Top: APs induced upon injection of 2 ms pulses of 1 nA in current-clamped neurons. Middle: Optical detection of APs with single trial measurement (S/N ~ 3). Bottom: increased S/N (~ 5) with averaging of five trials. (Pages et al, 2011)

We disagree however with the claim that “*the entire manuscript is based on optical recordings of a voltage dye*”. This is not true. As stated in our title, we combined two additional recording approaches, that unambiguously confirm the recordings made by the voltage sensitive dye. These were further confirmed through a pharmacological approach that we included in our additional experiments. This simultaneous triple recording aspect of the study is truly unique and provides internal controls that confirm each other mutually to verify our voltage signals. The experimental rigor that this novel method provides should not be discounted.

We also disagree with the claim that “*the exact chemical/optical properties of the dye in solution (or in HEK cells or in leech cells) are irrelevant.*” We cannot accept such a statement because it is simply wrong. The voltage sensitive dyes, ANNINE-6 and ANNINE-6plus, have been carefully developed over many years and rigorously tested to confirm their reliability for neuronal voltage imaging and as listed above, several independent groups have successfully used ANNINE-6plus in different cell types and preparations to confirm this.

Our approach has been different, and this is why we disagree that the exact chemical/optical properties of the dye in solution (or in HEK cells or in leech cells) are irrelevant: We designed the ANNINE dyes carefully and studied the physical chemistry and biophysical properties of the probe thoroughly. We proved that ANNINE dyes are purely electrochromic (Kuhn & Fromherz, JPhysChem, 2003). This means that the probe is independent of the membrane characteristics and the voltage sensitivity is based on a simple, linear, and fast mechanism. As such the dye works in all cell types the same way. Based on this result, we developed an imaging method to optimize the voltage response. The voltage response is excitation wavelength dependent and increases toward the red spectral edge of absorption (Kuhn, Fromherz, Denk, BiophysJ, 2004). In a previous paper we showed that this wavelength dependence holds in barrel cortex of the mouse (Fig 1g, Kuhn, Denk, Bruno, PNAS, 2007, “*In vivo two-photon voltage-sensitive dye imaging reveals top-down control of cortical layers 1 and 2 during wakefulness*”). Therefore, the probe behaves the same in vivo as in the leech neurons and the HEK293 cells and therefore the mechanism of voltage sensitivity is confirmed.

Since receiving the latest comments, we have now repeated these experiments on Purkinje neurons in vivo (see **supplementary Fig. S1e-d**) and explained in the text (see **line 83-90**). We show a similar wavelength dependence of ANNINE-6plus in Purkinje neurons in vivo. Again showing similar ANNINE-6plus behavior in all cell types, including Pyramidal neurons, leech neurons and HEK293 cells and now Purkinje neurons.

We can envisage no justifiable reason to suspect that ANNINE6-plus alters Purkinje neuron membranes any more than Pyramidal neuron membranes in vivo or in culture (Kuhn, Denk, Bruno, PNAS, 2007) (Pages et al. 2011) (Mennerick et al. 2010), or more than any other voltage sensitive dyes used on Purkinje neurons in vitro (Palmer et al. 2010), in fact the available evidence supports the opposite (Mennerick et al. 2010). We added these citations (see **line 78-79**).

Yet if there remains concerns that the voltage sensitive dye behaves differently or has an adverse physiological effect specific to Purkinje neurons. Our data also strongly indicates otherwise. Our in-vivo voltage recordings are supported by both calcium imaging and electrophysiology and are in agreement with previous dendritic patch-clamp recordings and calcium imaging experiments from Purkinje neurons in vivo. The fact that our labelled Purkinje neurons show normal physiological behavior in both simple spikes and complex spikes, using three independent recording techniques and that we can record voltage signals for up to two weeks after labelling the Purkinje neuron with ANNINE-6plus, proves its reliability and suitability for in vivo functional imaging. Whatever adverse effect ANNINE-6plus may have on neuronal function of Purkinje neurons, they remain undetectable for weeks after labelling the neuron. Furthermore, our

voltage signals responded as expected following pharmacological manipulation and recovered the following day, after drug washout.

Clearly such control experiments would be impossible to perform in slices, and yet these important controls are absent from all other studies using voltage sensitive dyes. Additionally, if we convert the relative fluorescence changes into voltage changes (based on measurements in leech neurons and HEK293 cells) we find remarkably similar voltage changes as reported with dendritic patch clamp experiments in Purkinje neurons in vivo. As such there is no reason to suspect unreliable voltage recording or adverse effects of ANNINE-6plus that is specific to Purkinje neurons.

3. If the authors do indeed demonstrate that the dye works as advertised, concerns remain about the significance of the scientific conclusions of this study. The authors have: 1 - Potentially “confirmed” some previous in vitro findings in awake animals. This is an insufficient advance in my view. And 2 - New pharmacological studies show that distal PN dendrites in awake animals have local spatio-temporal voltage fluctuations that are dependent on synaptic input. Of course they do. How could it work otherwise? This is only interesting if the authors were to observe the converse; that somehow dendritic spatio-temporal voltage is not dependent on synaptic input.

We disagree, as do many experts in the cerebellum and dendrite community, who have found our manuscript “super interesting” (citing from an email we received from cerebellum expert Prof. Samuel S-H Wang, Princeton University) and who were “very impressed” (citing from an email we received from cerebellum/dendrite expert Prof. Michael Häusser, University College London).

As noted previously we confirm many previous findings and thereby confirm our method. In addition, we also have several very exciting findings (also mentioned above) which would be impossible to achieve without the new method, these include:

- Discrete 1-2 ms supra-threshold voltage spikelets were detected during dendritic complex spikes in the distal spiny dendrites. Spikelets were highly heterogeneous in number, timing and spatial distribution within single complex spikes and between complex spikes. This heterogeneity was reflected by spatio-temporally variable dendritic calcium signals.
- Sub-threshold voltage imaging detected highly attenuated back-propagating action potentials, and spatio-temporal voltage maps revealed highly variable 5-10 ms sub-threshold ‘hotspots’ localized to fine dendritic processes, which were reduced in size and frequency by lidocaine and CNQX.
- “Hotspots” correlated with somatic output but also, at high frequency, triggered purely dendritic calcium spikes.

Relating specifically to the two points made in comment #3; 1 – How could the hotspot events we describe have been already found in an in-vitro preparation that lacks physiological synaptic input where the pattern of synaptic input is unknown? 2 – The fact that we prove expectations for

dendritic voltage fluctuation dependence on synaptic input, is a positive result and does not make the findings any less significant. Prior to our findings, an understanding of dendritic spatio-temporal membrane voltage fluctuations caused by synaptic input, and how they relate to somatic output was based purely on theoretical studies, which does not constitute a verifiable conceptual understanding of the process. This requires the experimental proof in vivo that we now provide.

4. Why are the authors only able to record electrically from their neurons for less than 1 min? Many groups routinely perform whole-cell patch-clamp recordings (which are much more difficult and more sensitive to movement than the cell-attached recordings used by the authors) during behavior in mice.

We can also hold the neuron much longer than 1 min extracellularly. However, in our case many other things have to coincide: The mouse must sit still (to avoid movement artifacts) and the line scan has to be perfectly positioned. If we slightly readjust the pipette position, the linescan position may change and the recordings repeated. The electrical recording is also done through an access port in a chronic cranial window which does not make the experiment easier, and frequently there are only a few labelled Purkinje neurons per mouse, which leaves very little room for trial and error. Everybody agrees, that these experiments are challenging.

5. Number of samples is not the issue – the length of time across which the sampling is conducted is what matters in this particular case. There is no compelling reason to sample GCaMP6 much more than 30 Hz based on the kinetics of the dye and the underlying Ca²⁺ signal. Of course, voltage is much faster and it must be sampled as such. What is the rationale for recording from dendritic segments for 500 s? Whatever it is, surely this should be enough to correlate dendritic signals with behavioral features (running, whisking, pupil dilation, posture, etc). Without any sort of behavioral result, it is still unclear what any of the recorded signals have to do with anything.

I appreciate that the authors feel that behavior is beyond the scope of the current manuscript, but I disagree, particularly in conjunction with the stated technical concerns. The methods component of the manuscript is not strong enough to carry a purely descriptive scientific result. Substantial, novel conclusions regarding general principles for dendritic integration and/or PN/cerebellar function in vivo during behavior would greatly strengthen this study.

We will absolutely attempt these experiments in future studies, by overcoming the technical obstacles involved. For completeness and rigor, we believe it is important (and more interesting) to begin by exploring in detail the complex and variable signaling that occurs in PNs during the

resting state of the animal, rather than immediately correlating activity with behavior. We also now provide in this study an important contrast with the signaling that occurs while the animal is under anesthesia.

Taking this approach, we aimed to explore the full potential and the limitations of our method and also provide to the scientific community a solid foundational understanding of PN function in vivo, upon which additional follow-up studies can be built.

In summary, the authors' approach has significant promise, but ground truth validation is necessary, as is correlation of dendritic signals with behavioral variables.

Thank you again for your critical feedback and thoughtful comments. We sincerely hope we were able to explain our thoughts and logic clearer than in our first rebuttal.